# Robot Policy Learning with Temporal Optimal Transport Reward

**Yuwei Fu**[1]    **Haichao Zhang**[2]    **Di Wu**[1]    **Wei Xu**[2]    **Benoit Boulet**[1]
[1]McGill University          [2]Horizon Robotics
`yuwei.fu@mail.mcgill.ca`    `haichao.zhang@horizon.cc`

## Abstract

Reward specification is one of the most tricky problems in Reinforcement Learning, which usually requires tedious hand engineering in practice. One promising approach to tackle this challenge is to adopt existing expert video demonstrations for policy learning. Some recent work investigates how to learn robot policies from only a single/few expert video demonstrations. For example, reward labeling via Optimal Transport (OT) has been shown to be an effective strategy to generate a proxy reward by measuring the alignment between the robot trajectory and the expert demonstrations. However, previous work mostly overlooks that the OT reward is invariant to temporal order information, which could bring extra noise to the reward signal. To address this issue, in this paper, we introduce the Temporal Optimal Transport (TemporalOT) reward to incorporate temporal order information for learning a more accurate OT-based proxy reward. Extensive experiments on the Meta-world benchmark tasks validate the efficacy of the proposed method. Our code is available at: `https://github.com/fuyw/TemporalOT`.

## 1   Introduction

Reinforcement Learning (RL) [51] has achieved great success across a wide array of applications [40]. However, it typically requires a large number of interactions with the environment [26, 35], which limits its practical application in the robotic control [9, 49]. A large body of work has been developed to address this issue from different aspects [52], *i.e.*, using curiosity-based intrinsic reward to encourage exploration [2, 47], leveraging better representation pretrained on large scale robotics datasets [33, 38], incorporating external knowledge from the Vision-Language Models (VLMs) [13, 32], and imitating the behaviors from pre-collected expert demonstrations [50, 62].

Reward specification plays a central role in RL [6]. Since the goal of the RL agent is to maximize the expected cumulative rewards, the reward signal directly influences the learned behaviors [8]. One major challenge in applying RL to real-world problems is how to design the reward functions [10]. A well-designed reward function can guide the agent towards desirable behaviors more efficiently, while a poorly designed one could lead to sub-optimal behaviors [25]. However, designing a good reward function is a nontrivial task [11], which requires related expert domain knowledge and (or) time-consuming hand reward engineering [48]. The lack of a good reward function is one of the main bottlenecks for the low sample-efficiency issue in RL [58].

Imitation Learning (IL) has been proven to be an effective technique to learn control policies without the oracle task reward [24]. Given an expert demonstration dataset, IL formulates the policy learning as a supervised learning paradigm [27]. The IL objective aims to learn a policy that mimics the expert behaviors via minimizing a distance measure of the learned policy and an approximated expert policy [23, 28]. Depending on if the RL agent can learn from further online interactions, IL can be crudely classified as offline IL and online IL [29]. In offline IL, the RL agent purely learns from a

38th Conference on Neural Information Processing Systems (NeurIPS 2024).

fixed dataset of collected expert experiences. In online IL, the RL agent usually learns a proxy reward function to relabel the collected online trajectories with respect to the expert demonstrations [53].

One notable weakness of IL is that it generally requires a diverse and high-quality demonstration dataset to achieve desired performances [5]. Recently, some researchers found that Optimal Transport (OT) [18, 44] based proxy reward function enables us to learn effective robot policies with only a few expert demonstrations [30, 31]. In this paper, we follow this line of research in applying OT-based proxy rewards to online IL without using any task reward information. In particular, we first revisit the efficacy of OT-based proxy reward in RL and then discuss some challenges of the existing methods due to the overlook of temporal order information. To mitigate this issue, we introduced the **Temporal O**ptimal **T**ransport (TemporalOT) reward, which incorporates temporal order information to the OT-based proxy reward via using context embeddings and a mask mechanism.

The primary contributions of this work can be summarized as follows:

- we pointed out a weakness of existing OT-based proxy reward methods for imitation learning due to the overlook of temporal order information;
- we designed a simple yet effective algorithm to incorporate temporal order information into OT-based proxy reward via using context embeddings and a mask mechanism;
- experiments show that the proposed method outperforms other SOTA algorithms.

## 2  Background

### 2.1  Reinforcement Learning

In this work, we consider the standard Markov Decision Process (MDP) [46] setting $\mathcal{M} = (\mathcal{S}, \mathcal{A}, R, P, \rho_0, \gamma)$, where $\mathcal{S}$ and $\mathcal{A}$ are state and action spaces, $R : \mathcal{S} \times \mathcal{A} \to \mathbb{R}$ is a reward function, $P : \mathcal{S} \times \mathcal{A} \to \Delta(\mathcal{S})$ is the state-transition probability function, $\rho_0 : \mathcal{S} \to \mathbb{R}_+$ is the initial state distribution and $\gamma \in [0, 1)$ is a discount factor. Our goal is to learn a policy $\pi(a|s) : \mathcal{S} \to \Delta(\mathcal{A})$ that maximizes the expected cumulative discounted rewards $\mathbb{E}_\pi[\sum_{t=0}^\infty \gamma^t r(s_t, a_t)]$ where $s_0 \sim \rho_0$, $s_{t+1} \sim P(\cdot|s_t, a_t)$ and $a_t \sim \pi(\cdot|s_t)$. In partially observable MDP (POMDP) [19], we can only receive an observation $o_i \in \mathcal{O}$, *i.e.*, image observation, of the current state $s_i$.

To solve this optimization problem, value-based RL methods typically learn a state-action value function $Q^\pi(s, a) := \mathbb{E}_\pi[\sum_{t=0}^\infty \gamma^t r_t | s_0 = s, a_0 = a]$, which is defined as the expected return under policy $\pi$. For convenience, we adopt the vector notation $Q \in \mathbb{R}^{\mathcal{S} \times \mathcal{A}}$, and define the one-step Bellman operator $\mathcal{T}^\pi : \mathbb{R}^{\mathcal{S} \times \mathcal{A}} \to \mathbb{R}^{\mathcal{S} \times \mathcal{A}}$ such that $\mathcal{T}^\pi Q(s, a) := r(s, a) + \gamma \mathbb{E}_{s' \sim P, a' \sim \pi}[Q(s', a')]$. The $Q$-function $Q^\pi$ is the fixed point of $\mathcal{T}^\pi$ such that $Q^\pi = \mathcal{T}^\pi Q^\pi$ [51]. Similarly, we define the optimality Bellman operator as follows $\mathcal{T}Q(s, a) := r(s, a) + \gamma \mathbb{E}_{s' \sim P}[\max_{a'} Q(s', a')]$ and the optimal $Q$-value function $Q^*$ is the fixed point of $\mathcal{T}Q^* = Q^*$. In deep RL, we use neural networks $Q_\theta(s, a)$ to approximate the $Q$-functions by minimizing the empirical Bellman error:

$$\mathbb{E}_{(s,a,r,s')} \left[ (r + \gamma \max_{a'} Q_{\hat{\theta}}^\pi(s', a') - Q_\theta^\pi(s, a))^2 \right], \tag{1}$$

where we sample transitions $(s, a, r, s')$ from a replay buffer and $Q_{\hat{\theta}}^\pi(s, a)$ is the target network.

### 2.2  Inverse Reinforcement Learning

Inverse Reinforcement Learning (IRL) aims to infer the underlying reward function from expert demonstrations [39], which further facilitates an RL agent to learn the policy. One key assumption of IRL is that the observed behaviors are optimal such that the observed trajectories maximize the cumulative rewards [17]. Due to the ability to avoid the manual reward specification, IRL holds the promise for practical real-world RL applications. Denote $\mathcal{M}$ as an MDP and $\pi^E$ as an expert policy, the IRL problem is to find an optimal reward function $R^*$ such that:

$$\mathbb{E}\left[\sum_{t=0}^\infty \gamma^t R^*(s_t, a_t) \big| \pi^E\right] \geq \mathbb{E}\left[\sum_{t=0}^\infty \gamma^t R^*(s_t, a_t) \big| \pi\right], \forall \pi \in \Pi, \tag{2}$$

where $\Pi$ is the feasible policy set. That is, the expert policy $\pi^E$ will achieve the maximum expected cumulative discounted reward than any other policy.

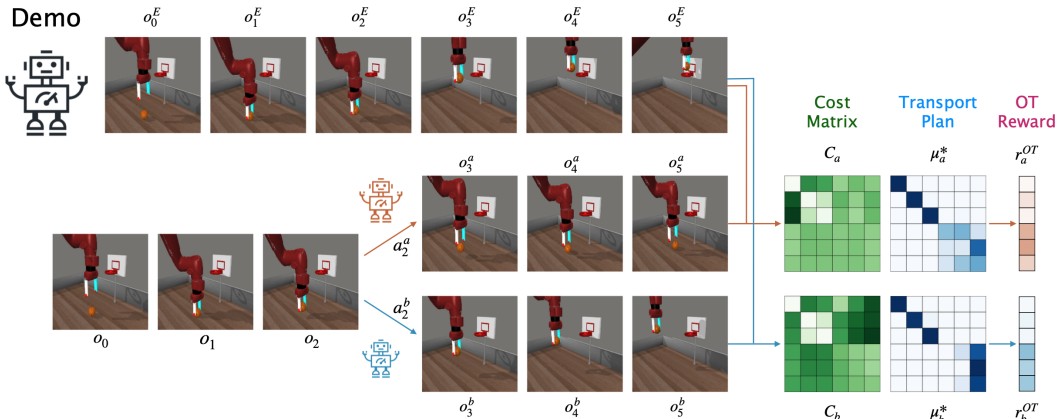

Figure 1: **An illustration of the pipeline of applying OT-based reward in RL.** In this toy example, we rollout two agent for five steps of transitions. Both agents start from the initial state and take same actions $a_0$ and $a_1$ at the first two states. Then the two agents take different actions $a_2^a$ and $a_2^b$ to generate different trajectories $\tau_a = (o_0, a_0, o_1, a_1, o_2, a_2^a, o_3^a, a_3^a, o_4^a, a_4^a, o_5^a)$ and $\tau_b = (o_0, a_0, o_1, a_1, o_2, a_2^b, o_3^b, a_3^b, o_4^b, a_4^b, o_5^b)$ The OT rewards for $(o_0, a_0)$ and $(o_1, a_1)$ in $\tau^a$ and $\tau^b$ are different even though the state-action pairs are exactly the same.

## 2.3 Optimal Transport

Optimal Transport is an optimization problem which aims to find an optimal mapping that transforms one probability distribution into another with the least cost. OT has a wide application in various domains such as economics [14], physics [15], and machine learning [1]. Consider two probability distributions $p \in \mathbb{R}^n$, $q \in \mathbb{R}^m$ and a joint distribution $\mu(p, q)$ on product space $\mathcal{X} \times \mathcal{Y}$, the Wasserstein distance [54] between $p$ and $q$ is defined as:

$$\mathcal{W}(p, q) = \inf_\mu \int_{\mathcal{X} \times \mathcal{Y}} c(x, y) d\mu, \tag{3}$$

where $c(x, y)$ is the cost function for moving mass form $x$ to $y$. In the RL scenario, $p$ and $q$ are usually in the state space $\mathcal{S}$ or the observation space $\mathcal{O}$ [21]. For example, given an expert trajectory $\tau^E = (o_1^E, \cdots, o_T^E)$ and an agent trajectory $\tau = (o_1, \cdots, o_T)$ where $o_i$ is the image observation at step $i$, the Wasserstein distance between $\tau^E$ and $\tau$ is defined in the following discrete form:

$$\mathcal{W}(\tau, \tau^E) = \min_{\mu \in \mathbb{R}^{T \times T}} \sum_{i=1}^{T} \sum_{j=1}^{T} c(o_i, o_j^E) \mu(i, j),$$

$$\text{s.t.} \sum_{i=1}^{T} \mu(i, j) = \sum_{j=1}^{T} \mu(i, j) = \frac{1}{T}. \tag{4}$$

where $\mu \in \mathbb{R}^{T \times T}$ is called the transport plan, and we denote the optimal transport plan as $\mu^*$.

## 3 Method

In this section, we first revisit the application of OT-based proxy reward in RL. In particular, we point out the influence of temporal order information, which has been overlooked in most prior work. Next, we introduce the main idea and formulation of the proposed method based on these observations.

### 3.1 A Recap of OT Reward in RL

#### 3.1.1 OT reward helps to rank states and actions

In RL, we usually adopt the Wasserstein distance to measure the similarity of two trajectories [31], as illustrated in Figure 1. Given an agent trajectory $\tau = (o_1, \cdots, o_T)$ and an expert trajectory

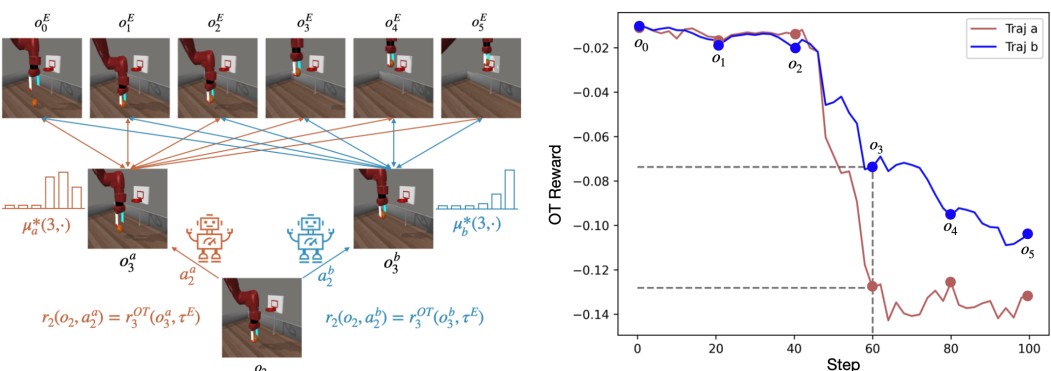

Figure 2: **Why OT reward could be useful?** When the OT reward is generally correct, it helps to rank the goodness of different states and induce the policy to take better actions. (left) In the toy example, two agents takes different action $a_2^a$ and $a_2^b$ at $o_2$ and thereafter. The goodness of $a_2^a$ and $a_2^b$ is measured by the OT reward computed w.r.t. to the observation of the next state $o_3^a$ and $o_3^b$. (right) A comparison of the true OT reward curves for trajectory $\tau^a$ and $\tau^b$, where $o_0/o_1/o_2/o_3/o_4/o_5$ correspond to observations at the 0/20/40/60/80/100-th step. We can observe that the OT reward for trajectory b is generally larger, which shows that the OT reward is generally correct.

$\tau^E = (o_1^E, \cdots, o_T^E)$, we first compute the optimal transport plan $\mu^*$ in Eqn.(4) using some iterative optimization algorithms, *i.e.*, Sinkhorn algorithm [4]. Then, the OT-based proxy reward at the $i$-th step is defined as follows:

$$r_i^{OT} = -\sum_{j=1}^{T} c(o_i, o_j^E)\mu^*(i, j), \tag{5}$$

where $c(o_i, o_j^E)$ is a cost function that measures the similarity between $o_i$ and $o_j^E$. One popular choice in prior work is the cosine similarity based cost function $c(o_i, o_j^E) = 1 - \frac{\langle f(o_i), f(o_j^E) \rangle}{\|f(o_i)\|\|f(o_j^E)\|}$, where $f(o_i)$ is the latent representation of observation $o_i$ extracted by a visual encoder [3].

Similar to the curiosity-based exploration bonus [2], OT reward is used to distinguish the goodness of different states. We consider the toy example in Figure 1, two agents start from the same state with observation $o_2$ and take different actions $a_2^a$ and $a_2^b$, respectively. Then the goodness of $a_2^a$ and $a_2^b$ at $o_2$ is measured by the OT reward computed w.r.t. the observation $o_3^a$ and $o_3^b$ at the next step. As long as the OT reward can rank $r_3^{OT}(o_3^a, \tau^E)$ and $r_3^{OT}(o_3^b, \tau^E)$ correctly, then the policy will be able to learn a better action at $o_2$. From Figure 2 (right), we can observe that the OT reward for the better trajectory b is generally larger, which validates the previous explanations.

### 3.1.2 Two Key Observations

Notably, there are two key observations of OT reward that have been less discussed in prior work:

1. The OT reward is order invariant.

2. The OT reward at step $i$ is influenced by the later steps so that two transitions with the same state-action pair could have different OT rewards.

Our first observation is that the standard OT-reward is order invariant. As shown in Eqn.(5), the order information is discarded and the frames from the demo trajectory are treated as bag-of-temporally-collapsed frames. In our view, collapsing the temporal axis drops arguably one of the most important characteristic features of temporal order information. More concretely, consider a demo trajectory of $\tau_1 = (o_1, o_1, o_2)$, meaning the agent first stays in the first state and then moves to the second state. Our goal is to imitate this behavior. However, if we discard the order information as in Eqn.(5), from the perspective of OT reward, there is no ability to differentiate between $\tau_1$ and some other undesired trajectories, *i.e.*, $\tau_2 = (o_1, o_2, o_1)$ which first moves to the second state and then moves back to the first state. Therefore, discarding the temporal order information in reward calculation

makes the reward on top of it under-constrained, thereby increasing the likelihood of convergence toward undesired solutions.

Our second observation is that the OT reward is non-stationary during the training. As the Eqn.(5) shows that OT reward $r_i^{OT}$ depends on the optimal transport plan $\mu^*(i, j)$, which depends on the entire trajectory. Therefore, the OT reward for each state is a function of the entire agent trajectory. As illustrated in Figure 1, even though trajectory $\tau^a$ and trajectory $\tau^b$ have the same first two transitions, their OT rewards have different values. This is very different from the standard RL setting, where reward is usually determined with a fixed state-action pair. Such a non-stationary OT reward could have a pitfall that makes the optimization of RL objective in Eqn.(1) to be less stable.

Inspired by these two observations, we found that the current OT-based RL methods usually overlook the temporal order information of the trajectories. In this work, we aim to investigate how to improve the current OT-based RL methods by incorporating temporal order information.

## 3.2 Temporal Optimal Transport Reward (TemporalOT)

In this subsection, we present the **Temporal Optimal Transport (TemporalOT)** reward. We first explain our motivations for the model design, and then introduce the details of the proposed method.

### 3.2.1 Motivation for the Model Design

The pipeline of a standard OT-based reward calculation usually consists of two stages:

(Stage-1) first define a transport cost function $c(\cdot, \cdot)$ between two states;

(Stage-2) then solve an OT optimization problem in Eqn.(4) to approximate the optimal transport plan $\mu^*$ and compute the OT reward $r^{OT}$ in Eqn.(5) for each state in a trajectory $\tau$.

After the OT-reward calculation step, the transition will be relabeled with the OT-reward for training an RL agent as in Eqn.(1). Our method aims to improve both stages of OT-reward calculation. Firstly, previous methods usually use a pair-wise cosine similarity based cost function in Stage-1, which sometimes could be inaccurate and noisy. Secondly, previous methods ignore the temporal order information in Stage-2 as discussed in Section 3.1.2. We will introduce two simple solutions to address these two points, respectively.

### 3.2.2 Context Embedding-based Cost Matrix for Improving Stage-1

To learn a more accurate transport cost function, we introduce a context embedding based cost matrix. Unlike previous methods that use a pair-wise cosine similarity as the transport cost, we adopt a group-wise cosine similarity that we define the transport cost between agent observation $o_i$ and expert observation $o_j^E$ as following:

$$\hat{c}(o_i, o_j^E) = \frac{1}{k_c} \sum_{h=0}^{k_c-1} \left( 1 - \frac{\langle f(o_{i+h}), f(o_{j+h}^E) \rangle}{\|f(o_{i+h})\| \|f(o_{j+h}^E)\|} \right), \tag{6}$$

where $k_c$ is the parameter of the context length and $f(\cdot)$ is a fixed visual encoder. The goal of the context cost matrix $\hat{C}$ is to facilitate expert progress estimation by taking nearby information into consideration. For example, we use $k_c = 3$ in Figure 3 and the transport cost between $o_1$ and $o_2^E$ is $\hat{c}(o_1, o_2^E) = 1 - [\cos(f(o_1), f(o_2^E)) + \cos(f(o_2), f(o_3^E)) + \cos(f(o_3), f(o_4^E))]/3$.

### 3.2.3 Temporal-masked Optimal Transport Objective for Improving Stage-2

The prevalent OT reward ignores the temporal order information and takes the information of every step in the trajectory into consideration, as shown in the Eqn.(5). As pointed out by some previous work, the OT reward is not always correct where noisy OT rewards could distract the agent from learning some key early behaviors [30]. To mitigate this issue, we introduce a concise solution by adding a temporal mask to the cost matrix.

For an agent trajectory $\tau = (o_1, \cdots, o_T)$ and an expert trajectory $\tau^E = (o_1^E, \cdots, o_T^E)$, we denote the context cost matrix as $\hat{C} \in \mathbb{R}^{T \times T}$ and the transport plan as $\mu \in \mathbb{R}^{T \times T}$. The row sum and column

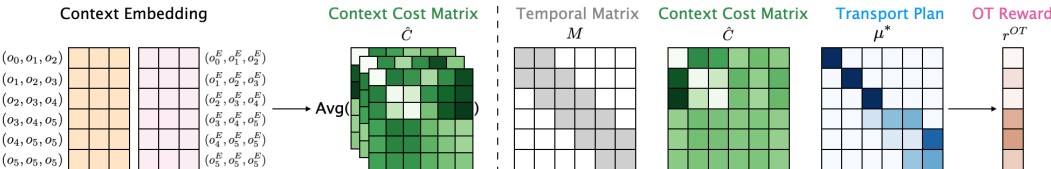

Figure 3: **An illustration of the proposed TemporalOT method.** (left) Instead of using a pair-wise cosine similarity as the transport cost, we use a group-wise cosine similarity to learn a more accurate cost matrix. (right) We use a temporal mask to enforce the OT reward to focus on a narrow scope to avoid potential distractions from observations outside of the mask window.

sum of $\mu$ equal to the constraint $\mathbf{s} = [\frac{1}{T}, \cdots, \frac{1}{T}] \in \mathbb{R}^T$. We proposed to introduce a temporal mask $M \in \mathbb{R}^{T \times T}$ to the transport plan, where $M(i,j) \in [0,1]$. We can express the masked optimal transport objective in the following vector form [16]:

$$\mu^* = \arg\min_{\mu} \langle M \odot \mu, \hat{C} \rangle_F - \epsilon \mathcal{H}(M \odot \mu), \text{ s.t. } \mu\mathbf{1} = \mu^T\mathbf{1} = \mathbf{s}, \tag{7}$$

where $\langle \cdot, \cdot \rangle_F$ is the Frobenius norm and we add an entropy regularizer $\mathcal{H}(\cdot)$ of the masked transport plan $M \odot \mu$. We can solve Eqn.(7) by the Lagrangian:

$$L(\mu, \alpha, \beta) = \langle M \odot \mu, \hat{C} \rangle_F + \epsilon \left( \langle M \odot \mu, \log(M \odot \mu) \rangle_F - \mathbf{1}^T(M \odot \mu)\mathbf{1} \right) - \tag{8}$$
$$\langle \alpha, (M \odot \mu)\mathbf{1} - \mathbf{s} \rangle_F - \langle \beta, (M \odot \mu)^\top\mathbf{1} - \mathbf{s} \rangle_F,$$

where $\alpha$ and $\beta$ are two Lagrangian multipliers. By using different temporal mask, we can control what kind of temporal order information we use in the OT reward. For example, $M = \mathbf{1}$ degrades to the original OT reward without temporal order information, and a lower triangle matrix corresponds to the causal mask in the Transformer decoder [55], which indicates that we only concern the past steps observations. In our method, we use a variant of the diagonal matrix:

$$M(i,j) = \begin{cases} 1, & \text{if } j \in [i - k_m, i + k_m], \\ 0, & \text{otherwise}, \end{cases} \tag{9}$$

where $k_m$ is a window size parameter that controls the scope we use in the masked OT rewards. A smaller mask window size $k_m$ refers to a closer match w.r.t. to the expert demonstration. We select a diagonal-like matrix because we follow previous learning from demonstration literature to assume that the agent has a similar movement speed as the expert [30]. Under this assumption, we adopt the distance between the time step indexes to represent temporal affinity information. Figure 3 illustrates the main ideas of the proposed TemporalOT method.

## 4 Experiments

In this section, we aim to answer the following questions: (1) How does the proposed TemporalOT method perform compared with other baselines? (2) Are the proposed context-embedding based cost function and temporal mask useful? (3) How do the key parameters influence the performances? (4) Is TemporalOT effective with both state-based and pixel-based observations?

### 4.1 Experimental Setup

We implement TemporalOT-RL in PyTorch [42] based on the official ADS implementation[1]. We use a pretrained ResNet50 [20] network as the fixed visual encoder to extract the image embedding for each pixel observation. Unlike the original ADS experiments which use a fixed goal in each task, we adopt a more challenging setting where the goal position changes for each episode. Moreover, we only provide two expert video demonstrations to the RL agent. For the experiment results, we evaluate the RL agent for 100 trajectories every 20000 steps. We report the mean and standard deviation of the evaluation success rate across 5 random seeds. We define one trajectory to be successful if the RL agent solves the task at the last step. More detailed information is available in the Appendix B.

---

[1] https://github.com/dwjshift/IL_ADS

Table 1: **Experiment results of success rate on the Meta-world benchmark.**

| Environment | TaskReward | BC | GAIfO | OT0.99 | OT0.9 | ADS | **TemporalOT** |
|---|---|---|---|---|---|---|---|
| Basketball | 0.0 (0.0) | 0.2 (0.4) | 0.0 (0.0) | 0.0 (0.0) | 76.6 (27.4) | 42.2 (44.5) | **94.4 (4.7)** |
| Button-press | 14.0 (18.5) | 1.7 (2.4) | 1.0 (1.1) | **88.8 (2.5)** | 85.2 (3.3) | **89.0 (3.8)** | 92.4 (3.6) |
| Door-lock | 86.2 (12.4) | 4.6 (7.0) | 8.8 (12.2) | 3.0 (5.5) | 2.8 (2.0) | 3.2 (2.7) | **33.4 (2.8)** |
| Door-open | 0.0 (0.0) | 10.7 (10.3) | 2.2 (1.7) | 46.2 (33.6) | 30.2 (34.5) | 52.0 (42.7) | **78.4 (12.4)** |
| Hand-insert | 0.8 (1.6) | 2.3 (2.1) | 8.6 (4.4) | 29.0 (9.7) | 11.2 (2.3) | **35.0 (5.3)** | 36.8 (6.6) |
| Lever-pull | 0.0 (0.0) | 0.8 (1.6) | 3.4 (1.9) | 15.4 (15.5) | 35.6 (12.8) | 21.2 (12.0) | **53.6 (7.7)** |
| Push | 1.0 (0.7) | 0.4 (0.8) | 0.0 (0.0) | **14.2 (7.5)** | 7.0 (2.6) | **17.2 (5.6)** | 8.4 (1.7) |
| Stick-push | 0.0 (0.0) | 0.0 (0.0) | 18.8 (22.9) | 0.0 (0.0) | 48.8 (41.5) | 20.0 (40.0) | **97.6 (2.6)** |
| Window-open | 85.6 (12.2) | 1.6 (2.7) | 4.0 (4.7) | **54.0 (28.0)** | 22.4 (22.9) | 43.6 (20.5) | **55.2 (2.3)** |
| Average | 20.8 | 2.5 | 5.2 | 27.8 | 35.5 | 35.9 | **61.1** |

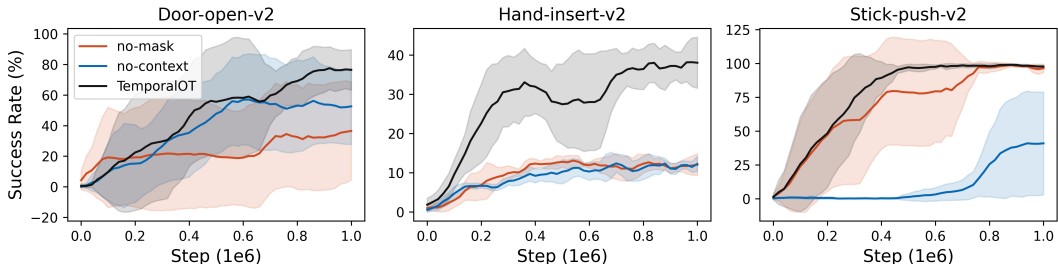

Figure 4: **Ablation for model components.** Both proposed components are useful.

## 4.2 Baselines

We compare the following baseline methods. (1) TaskReward: training a backbone RL agent from DrQ-v2 [56] with the oracle task reward $r_i^{task} = \delta_{success}$. The reward is 1 when the task is solved and otherwise the reward is 0. (2) BC: a naive behavior cloning agent which has the access of the expert action. (3) GAIfO: another IL baseline which learns a discriminator to provide proxy reward [53]. (4) OT: we use an online version of OTR [31] agent where we first rollout the RL agent to collect online trajectories and then relabel the reward with OTR for RL training. (5) ADS: a variant of OT baseline which adaptively adjusts the discount factor w.r.t. a progress tracker [30].

## 4.3 Results on the Meta-world Benchmark Tasks

We first validate the effectiveness of TemporalOT on nine Meta-world [57] tasks. Experiment results are shown in Table 1. We can observe that TemporalOT generally outperforms the other baselines without using the task rewards. Moreover, the TaskReward baseline only shows good performance on the *Door-lock* and *Window-open* tasks. The main reason is that the RL agent fails to collect the first successful trajectory. For example, the goal of the *Basketball* task as shown in Figure 1 is to pick up the basketball and move to a target position above the rim. With the oracle sparse task reward, the RL agent only receives a nonzero reward until it first successfully solves the task. Under such circumstances, it is particularly challenging to collect the first successful trajectory with only zero task rewards and random action explorations. On the other hand, we can observe that the two IL baselines, BC and GAIfO [53], perform much worse than other OT-reward based baselines. This is because we only provide two expert video demonstrations with a few hundred samples, where the IL-based methods suffer from an over-fitting issue. We further compare two OT agent baselines, where OT0.99 uses $\gamma = 0.99$, and OT0.9 uses $\gamma = 0.9$. We have a similar conclusion as in ADS that using a smaller discount factor is helpful to learn early behaviors in some tasks that strongly rely on the progress dependency, *i.e.*, *Basketball*. TemporalOT outperforms the recent SOTA baseline ADS in 8 out of 9 tasks, which proves the effectiveness of the proposed method.

## 4.4 Ablation Studies on Different Model Components

We further conduct ablation studies to validate the effectiveness of the proposed context cost matrix and temporal mask in TemporalOT. In Figure 4, *no-mask* refers to a variant of TemporalOT without mask and *no-context* refers to a variant of TemporalOT without the context cost matrix. We can

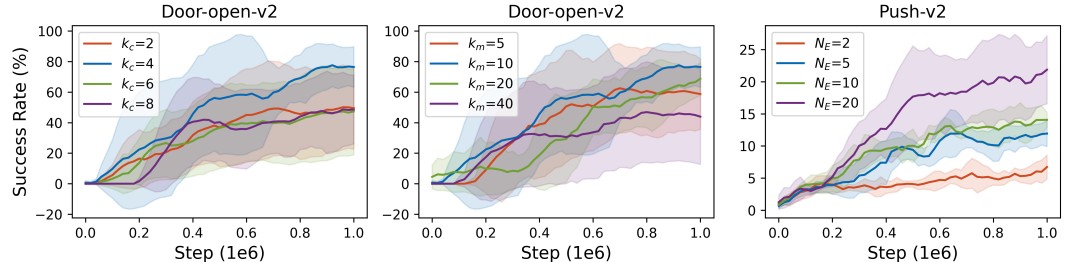

Figure 5: **Influences of key parameters.** A medium number of context length $k_c$ or mask length $k_m$ performs the best. The agent performs better with more expert demonstrations.

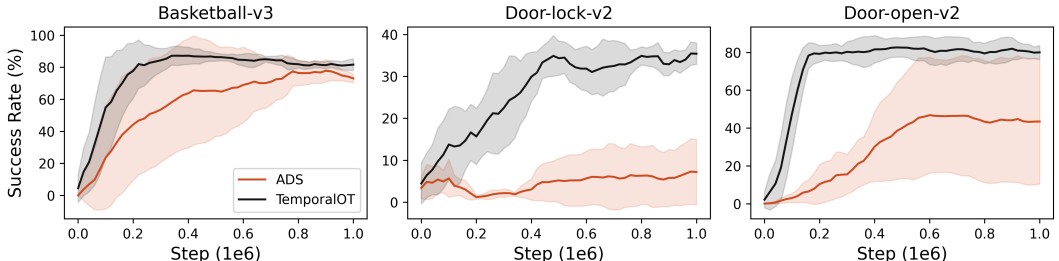

Figure 6: **Results with pixel-based inputs.** TemporalOT is also effective with pixel-based inputs.

observe that removing any of the two components will lead to a degraded performance. Moreover, the more important component varies depending on the task. For example, the temporal mask is more important in the *Door-open* task, and the context cost matrix is more important in the *Stick-push* task.

## 4.5 Ablation Studies on Different Key Parameters

We then validate the efficacy of different key parameters, *i.e.*, the context length $k_c$ for the context embedding, window size $k_m$ for the temporal mask, and the demonstration number $N_E$. From Figure 5, we can observe that a medium number of $k_c$ and $k_m$ performs the best and a larger $N_E$ improves the performances. A large context length $k_c$ does not perform well because it will distract the OT reward from the current step and introduce extra reward noise. A smaller mask window size $k_m$ makes the learning more difficult because it only receives information from nearby observations, and a larger $k_c$ will gradually degrade to the naive OT reward. Further, having more expert demonstrations is helpful in mitigating the potential over-fitting issue and improving the final performance.

## 4.6 Results with Pixel-based Observations

We also evaluate the proposed method with pixel-based observations, where we follow the same DrQ-v2 model setting as the ADS baseline. Figure 6 shows the results of the comparison of TemporalOT with ADS. We can observe similar conclusions as in Table 1 that our proposed TemporalOT method also outperforms the ADS baseline with pixel-based inputs, where TemporalOT usually converges faster than ADS and (or) achieves a higher final success rate. Moreover, we can observe that sometimes the pixel-based agent learns faster than its dense state-based counterpart, which indicates that the agent can extract more effective representations from the pixel inputs.

## 4.7 Visualization for Bad Cases

In this subsection, we visualize some bad cases of OT-based RL agents to provide readers more insights about when OT-based RL agents are less useful. Figure 7 plots a typical bad case for the OT/ADS/TemporalOT agents in the *Hand-insert* task. The top row is the expert trajectory, and the second row is the agent trajectory. The goal of the *Hand-insert* task is to move the brown block to a target position in the hole. We can observe that the RL agent mainly focuses on imitating the arm behaviors which ends in the target position, but it ignores the brown block. The main reason for the this bad case is that the color of brown box is very close to the table background which sometimes

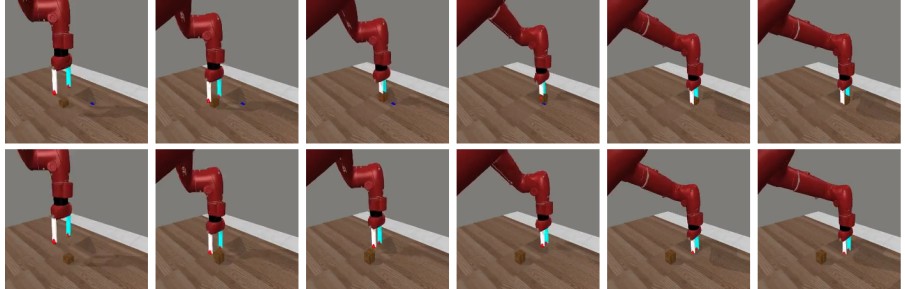

Figure 7: **Bad case analysis.** Compared with the expert trajectory (top), the agent focused on imitating the arm behavior (bottom) and missed the details, *i.e.*, grasping the block.

make it difficult for the pretrained visual encoder to capture the subtle information. More bad case analyses are available in the Appendix A.2.

## 5  Related Work

**Learning with a Few Demonstrations.** There is a large body of work on leveraging demonstrations for policy learning, ranging from the basic behavior cloning [12, 45] to demonstration-aided RL [43]. There is also work on leveraging demonstration data for offline pre-training [60, 36, 61], to either warm-start the policy [36] or help with exploration [61, 37, 22]. However, the amount of demonstrations required for a high-quality pre-training is typically large. In this work, we focus on the setting where only a small number of demonstrations are provided [5], thus greatly relieving the burden of generating demonstrations. Optimal Transport based imitation is a recently emerged approach in this direction, which will be reviewed in the subsequent section.

**Optimal Transport-based Reward for Imitation and RL.** Optimal Transport (OT) has been shown to be effective for imitation learning [18, 31]. Optimal Transport Reward Labeling (OTR) [31] uses Sinkhorn distance [4] to compute a similarity metric for a trajectory w.r.t. an expert demonstration and uses this metric as rewards for offline RL datasets without rewards [31]. Automatic Discount Scheduling (ADS) [30] uses a similar OT-based approach for reward calculation. The core idea of ADS is to incorporate a scheduling of the discount factor for online RL to mitigate the potentially distracting OT reward from temporally distant states. Our work aligns with previous work in this category, and addresses some common issues that are shared by previous methods.

## 6  Limitations

Since our work is closely related to IL, our method shares some common limitations of IL. For example, the success of our method heavily depends on high-quality expert video demonstrations. If we are facing a new task without any available expert demonstrations, our method will be less useful. Moreover, if the given demonstrations are sub-optimal or biased, the learned policies will inherit these flaws as well. Moreover, the performance of the proposed method relies on the quality of the pretrained visual encoder. If the pretrained visual encoder fails to capture some key information in the pixel observation, then our method will also fail to take such key details into consideration. Another limitation of our work is that the computation cost of the proposed method is related to the number of the given expert video demonstrations. A larger number of expert demonstrations will increase the computation cost when we compute the optimal transport plan.

## 7  Conclusion

This paper studies the problem of learning effective robot policies with expert video demonstrations. We focus on a challenging setting where there are only two demonstrations available and the environment does not provide any task reward. Following the line of research of OT-based proxy reward, we first discuss some challenges of the existing methods due to the overlook of temporal information. Further, we introduced a new method named TemporalOT, which incorporates temporal information to existing baseline by using a context-embedding based cost matrix and a mask mechanism. Experiments on nine Meta-world benchmark tasks showcase the effectiveness of the proposed method. One interesting future direction is to extend the current method to a camera-view invariant agent, where we can learn policies w.r.t. expert video demonstrations from different camera views.

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

# Appendix

# A    Additional Experiment Results

## A.1    Evaluation Curves

Figure 8 shows the evaluation curves corresponding to the results in Table 1. We can observe that TemporalOT generally outperforms the other baselines which do no use the task reward.

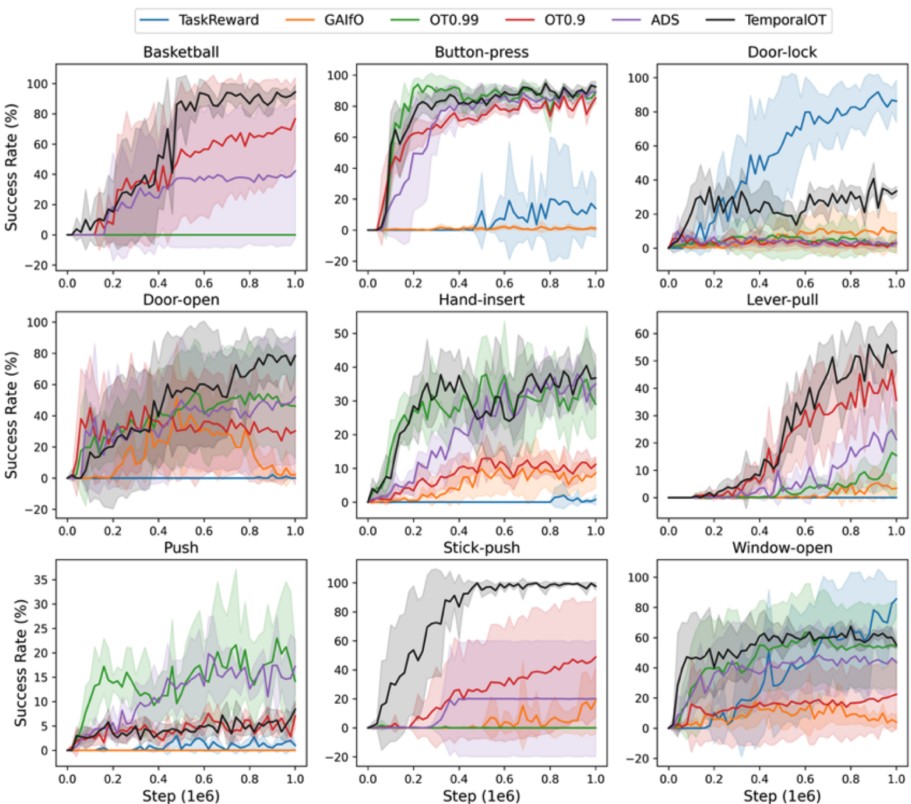

Figure 8: Evaluation curves on the Meta-world benchmark tasks.

## A.2    Bad Case Analysis

In this subsection, we analyze more bad cases in different tasks. Similar to Figure 7, we visualize the expert trajectory (top) and an agent trajectory (bottom) of some typical bad cases in different tasks. We can observe that the robot arm generally displays a similar behavior as demonstrated in the expert trajectory. The agent did not solve these tasks because it – (A) failed to grab the handle in the *Door-open* task; (B) failed to touch the knob in the *Door-unlock* task; (C) failed to pick the red block in the *Push* task; (D) failed to pick up the blue stick in the *Stick-push* task.

## A.3    Pretraining with Expert Data

In this subsection, we validate the effectiveness of using imitation learning, *i.e.*, behavior cloning (BC), to first initialize the robot policy and then fine-tune it with TemporalOT. In this experiment, we use action-inclusive expert demonstrations to pretrain the robot policy with behavior cloning loss. Table 2 shows the results on the *Door-open* task. The BC baseline is a pure offline method where the parameters are fixed after pretraining. TemporalOT-P is the variant that fine-tunes a pretrained BC policy using TemporalOT. We can observe that incorporating pretraining helps to improve the sample efficiency. Moreover, we can notice a small success rate drop (which recovers later) at the initial phase when we transit from offline to online training from step 0 to step 4e4. This is an *initial-dipping*

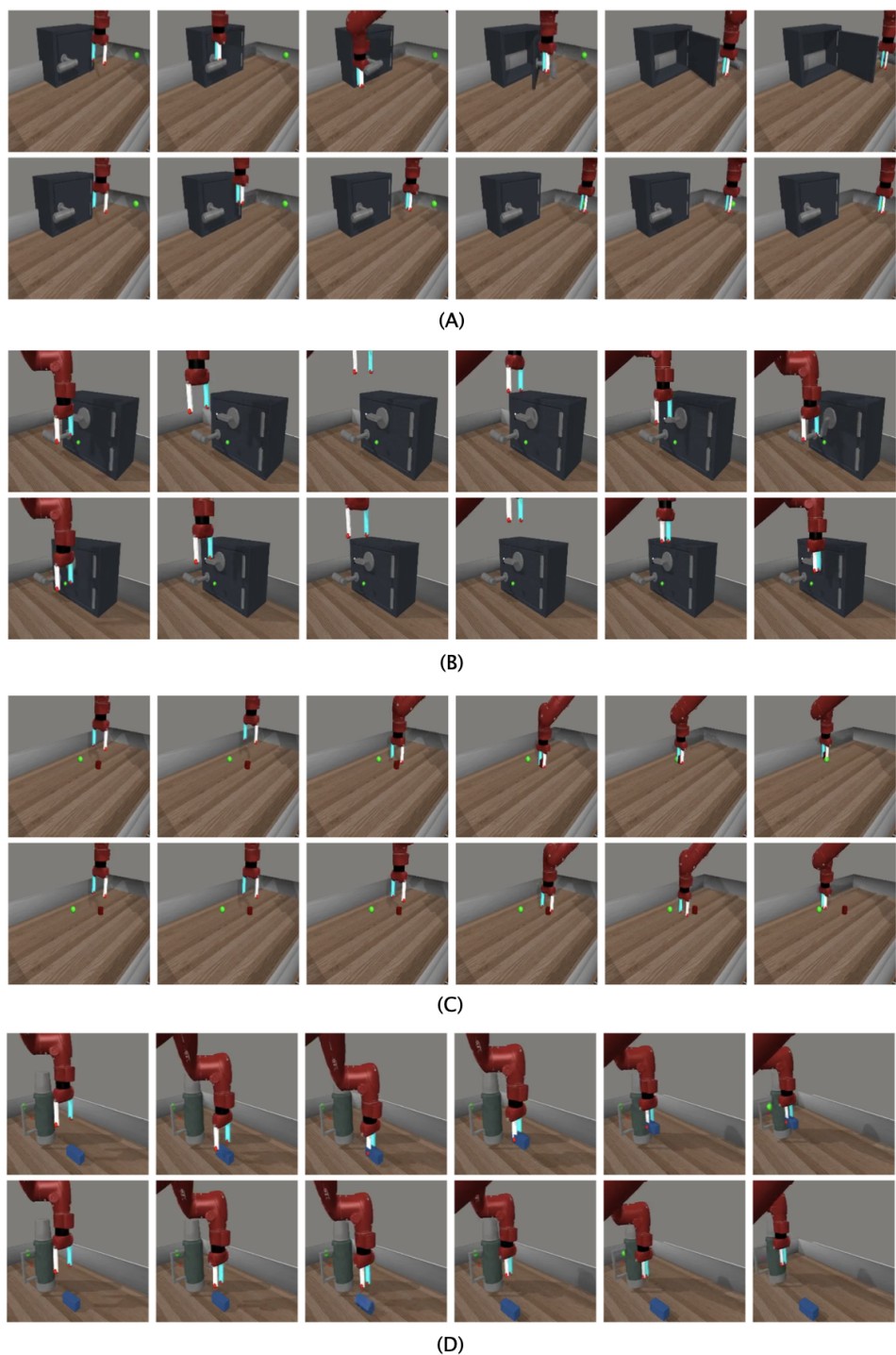

Figure 9: Visualization of more bad cases in the (A) *Door-open* task, (B) *Door-unlock* task, (C) *Push* task and (D) *Stick-push* task.

phenomenon as also being reported in previous offline RL literature and can be improved by designing more specific offline-to-online RL methods [59, 60].

Table 2: **Pretraining with action-inclusive demonstrations is helpful.**

|  | 0 | 2e4 | 4e4 | 6e4 | 8e4 | 1e5 | 5e5 | 1e6 |
|---|---|---|---|---|---|---|---|---|
| BC | 10.8 | - | - | - | - | - | - | - |
| TemporalOT | 0 | 2.0 | 0 | 0 | 5.0 | 16.6 | 57.8 | 78.4 |
| TemporalOT-P | 10.8 | 6.8 | 25.0 | 42.8 | 48.6 | 55.4 | 70.8 | 82.0 |

Table 3: **Experiment results of success rate for demonstrations with different speed.**

|  | 1x | 2x | 3x | 4x |
|---|---|---|---|---|
| Basketball | 94.4 (4.7) | 91.8 (8.9) | 77.2 (8.4) | 43.8 (29.5) |
| Button-press | 89.0 (3.8) | 72.4 (3.0) | 65.6 (5.4) | 57.8 (7.4) |
| Door-open | 78.4 (12.4) | 76.2 (10.4) | 49.4 (20.3) | 20.4 (9.2) |

## A.4 Ablation Studies for Expert Demonstration with Different Speed

Our work inherits an implicit assumption from learning from demonstration literature that the agent has a similar movement speed as the expert agent. Under this assumption, we use the distance between the time step indexes to represent temporal affinity information in the temporal mask. If the discrepancy between expert-agent movement speed is large, then the assumption will be broken. To validate how does such discrepancy influence the model performance, we run experiments using expert demonstrations with different speed. We generate an N times faster expert demonstration by sampling the original expert trajectory every N steps. Table 3 shows the results of using expert demo with different speed. We can observe that as the discrepancy between the expert-agent movement speed increases, the final success rate decreases. Using a double speed demonstration in the *Basketball* task and *Door-open* task achieves a slightly worse performance than the original demonstration. However, using a three times faster or four times faster expert demonstration performs quite badly. This is because a faster expert trajectory corresponds to a shorter sequence of demonstration frames, where it is more likely that the important information is dropped.

## A.5 Ablation Studies for Different Visual Encoders

In this subsection, we compare the effectiveness of using different visual encoders in TemporalOT. In particular, we compare three different ResNet variants (ResNet18, ResNet50, ResNet152) using the checkpoints from torchvision [34]. From Figure 10 (left), we can observe that ResNet50 and ResNet152 encoders show similar final performances. Here, ResNet18 underperforms the other two encoders because it is quite weak, such that sometimes it fails to capture the key information of the image. The results in the Figure 10 (left) indicate that a reasonably good visual encoder is usually enough to extract effective visual embeddings for computing OT rewards in RL.

## A.6 Ablation Studies for Different Mask Designs

In this subsection, we try a variant of TemporalOT that uses a dynamic mask in computing the OT rewards. Unlike Eqn.(9) where the masked position $M(i, j)$ always follows a fixed rule that $j \in [i - k_m, i + k_m]$, we introduce the following dynamic counterpart,

$$M(i, j) = \begin{cases} 1, & \text{if } j \in [c - k_m, c + k_m], \\ 0, & \text{otherwise,} \end{cases} \tag{10}$$

where the mask window center $c = \arg\min_j \hat{C}(i, j)$ and $j \in [0, i]$. Eqn.(10) means we select an index $j$ in the expert trajectory that has the lowest transport cost w.r.t. the current observation $o_i$. We further add a constraint $j \in [0, i]$ to avoid looking into distant future steps as pointed in Section 3.2.3. The experiment results in Figure 10 (right) show that the learning-based temporal mask in Eqn.(10) slightly outperforms our previous rule-based temporal mask in Eqn.(9). Since the main focus of this work is to use a simple and easy to understand design to illustrate the benefits of incorporating temporal information into OT rewards in RL, we leave the investigation of more sophisticated dynamic temporal masks to the future work.

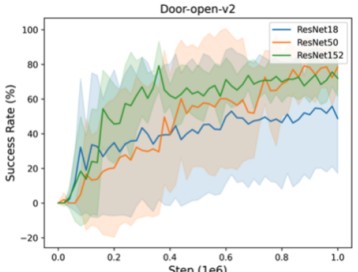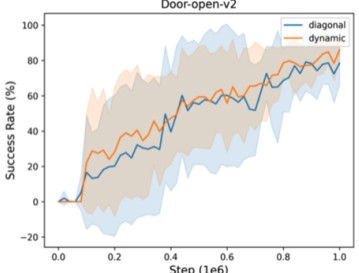

Figure 10: **More ablation studies.** (left) A comparison of different visual encoders. (right) Using a dynamic temporal mask slightly improves the performance.

## B   Experimental Setup

### B.1   Evaluation Environments

In the experiments, we focus on the Meta-world benchmark tasks [57]. Figure 11 shows the nine selected environments that we use in the experiments. In each environment, the robot arm aims to solve a specific task. We use the same task length parameter from the ADS paper [30]. The task lengths for the *Basketball* task and *Lever-pull* task are 175 steps and the task lengths for the other tasks are 125 steps. The goals for the nine selected tasks are as following:

- Basketball task: the arm aims to grasp the orange basketball and move to a target position above the rim.
- Button-press task: the arm aims to push down the red button.
- Door-lock task: the arm aims to rotate the knob to a target angle.
- Door-open task: the arm aims to open the door to a target position.
- Hand-insert task: the arm aims to move a brown block to a target position in the hole.
- Lever-pull task: the arm aims to move the lever to a target height.
- Push task: the arm aims to move the red cylinder to a target position on the table.
- Stick push task: the arm aims to grab the blue stick and push the bottle to a target position.
- Window open task: the arm aims to open the window to a target position.

### B.2   Implementation Details

Our code is based on the official implementation of ADS in PyTorch [41]. We also use the Meta-world environment [57] provided in the ADS codebase. For the fixed pretrained visual encoder, we use the Resnet50 [20] network trained on the ImageNet dataset [7]. In particular, we adopt the official Resnet50 checkpoint provided by Torchvision [34]. For the other main softwares, we use the following package versions:

1. Python 3.9.19
2. numpy 1.26.4
3. torch 2.2.2
4. torchvision 0.17.2
5. pot 0.9.3
6. dm-control 1.0.17
7. dm_env 1.6
8. mujoco-py 2.1.2.14
9. cython 3.0.0a10
10. gym 0.22.0

The pseudo-code of TemporalOT is summarized in Algorithm 1.

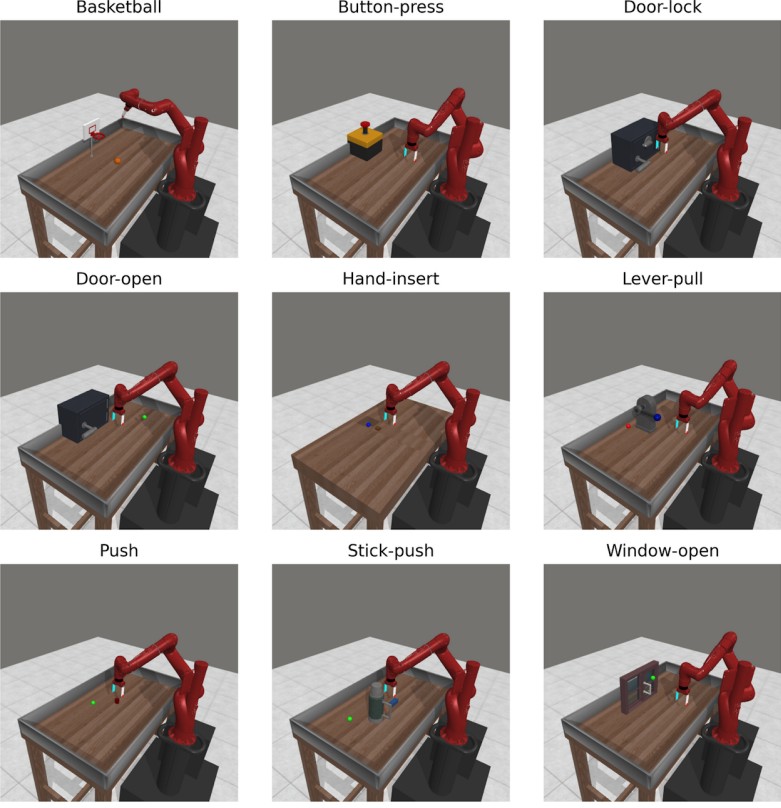

Figure 11: Visualization of the nine evaluation tasks in the experiments.

---

**Algorithm 1** Temporal Optimal Transport Reward (TemporalOT)

---

**Input:** a fixed pretrained visual encoder $f$, $N_E$ expert video demonstrations, temporal mask window size $k_m$, context embedding length $k_c$, trajectory length $T$, total trajectory number $N$, experience replay buffer $\mathcal{D}$.
**Output:** trained RL agent $\pi(a|s)$.
**for** $i = 1$ **to** $N$ **do**
    Unroll policy $\pi(a|s)$ to collect a trajectory $\tau = (o_1, \cdots, o_T)$.
    Compute visual embeddings using of the trajectory $(f(o_1), \cdots, f(o_T))$.
    **for** $j = 1$ **to** $N_E$ **do**
        Compute the context cost matrix $\hat{C}$ as in Eqn.(6) for the $j$-th expert demonstration.
        Compute the masked OT reward $R_\tau^j = (r_0^j, \cdots, r_T^j)$ with Eqn.(9).
    **end for**
    Select the expert demonstration with the largest trajectory OT reward sum as the final OT reward $r_\tau^{OT} = (r_0^{OT}, \cdots, r_T^{OT})$.
    Save the labeled transition $(o_i, a_i, r_i^{OT}, o_{i+1})$ to the replay buffer.
    Update the RL agent with sampled transitions as in Eqn.(1).
**end for**

---

## B.3 Parameter Settings

In the experiments, we mainly follow the parameter settings as in the ADS baseline. Some of the key parameters are summarized in the Table 4.

Table 4: Summarization of hyper-parameters.

| Parameter | Value |
|---|---|
| Total environment step | 1e6 |
| Adam learning rate | 1e-4 |
| Batch size | 512 |
| Target network $\tau$ | 0.005 |
| Discount factor $\gamma$ | 0.9 |
| Expert demo number $N_E$ | 2 |
| Context length $k_c$ | 3 |
| Mask window size $k_m$ | 10 |
| DrQ buffer size | 1.5e5 |
| DrQ action repeat | 2 |
| DrQ frame stack | 3 |
| DrQ image size | (84, 84, 3) |
| DrQ embedding dimension | 50 |
| DrQ CNN features | (32, 32, 32, 32) |
| DrQ CNN kernels | (3, 3, 3, 3) |
| DrQ CNN strides | (2, 1, 1, 1) |
| DrQ CNN padding | VALID |
| DrQ actor network | (1024, 1024, 1024) |
| DrQ critic network | (1024, 1024, 1024) |

### B.4 Computation Resources

We run our experiments on a workstation with an NVIDIA GeForce RTX 3090 GPU and a 12th Gen Intel(R) Core(TM) i9-12900KF CPU. The average wall-clock running time for TemporalOT on the state-based experiment and pixel-based experiment are 1 hour and 3 hours, respectively.

## C  Solution of the Masked OT Objective

In this section, we provide the solution of the masked OT objective in the Eqn.(7).

$$\mu^* = \arg\min_{\mu} \langle M \odot \mu, C \rangle_F - \epsilon \mathcal{H}(M \odot \mu), \text{ s.t. } \mu \mathbf{1} = \mu^T \mathbf{1} = \mathbf{s},$$

To compute the optimal transport plan $\mu*$, we first write its Lagrangian as follows:

$$L(\mu, \alpha, \beta) = \langle M \odot \mu, C \rangle_F + \epsilon \left( \langle M \odot \mu, \log(M \odot \mu) \rangle_F - \mathbf{1}^T (M \odot \mu) \mathbf{1} \right) - \langle \alpha, (M \odot \mu) \mathbf{1} - \mathbf{s} \rangle_F - \langle \beta, (M \odot \mu)^\top \mathbf{1} - \mathbf{s} \rangle_F.$$

We set the partial derivative of the Lagrangian to zero:

$$\frac{\partial L}{\partial \mu_{i,j}} = M_{i,j} C_{i,j} + \epsilon M_{i,j} \log(M_{i,j} \mu_{i,j}) - M_{i,j} \alpha_i - M_{i,j} \beta_j = 0.$$

If $M_{i,j} = 1$, then we have:

$$\log(\mu_{i,j}) = \frac{\alpha_i}{\epsilon} + \frac{\beta_j}{\epsilon} - \frac{C_{i,j}}{\epsilon}.$$

We can express it in the matrix form:

$$M \odot \mu = \text{diag}(\mathbf{u}) K \text{diag}(\mathbf{v}),$$

where $\mathbf{u} = e^{\alpha/\epsilon}$, $K = M \odot e^{-C/\epsilon}$, and $\mathbf{v} = e^{\beta/\epsilon}$. The constraints can be expressed as:

$$\text{diag}(\mathbf{u}) K \text{diag}(\mathbf{v}) \mathbf{1} = (\text{diag}(\mathbf{u}) K \text{diag}(\mathbf{v}))^\top \mathbf{1} = \mathbf{s}.$$

We can later use the Sinkhorn algorithm [4] to iteratively compute $\mathbf{u}$ and $\mathbf{v}$:

$$\mathbf{u}^{(l+1)} = \frac{\mathbf{s}}{K \mathbf{v}^{(l)}},$$

$$\mathbf{v}^{(l+1)} = \frac{\mathbf{s}}{K^\top \mathbf{u}^{(l+1)}}.$$

# D   Broader Societal Impacts

In this work, we investigate the application of OT-based proxy reward in learning effective robot policies with a few expert video demonstrations. Since there are usually no available task rewards in real-world scenarios, our method holds promise for the advancement of real-world RL applications. On the other hand, the proposed method aims to learn a policy that behaves similarly to the given demonstrations. Once the given expert demonstration contains some dangerous behaviors, our method is also likely to learn the dangerous behaviors, which will lead to some negative societal impacts. To mitigate such potential negative societal impacts, we can introduce an extra safety reward to prevent the agent from learning dangerous behaviors.

