# OpenReview forum: "Robot Policy Learning with Temporal Optimal Transport Reward"
_NeurIPS.cc/2024/Conference — NeurIPS 2024 poster_

### Official Review · Reviewer_GZSS · 2024-07-04

**Soundness:** 2
**Presentation:** 2
**Contribution:** 2
**Rating:** 5
**Confidence:** 4

**Summary:**

This paper extends the vanilla optimal transport-based proxy reward method in imitation learning by 1) masking out distant steps within a trajectory in the optimization of the transport plan and 2) considering neighbouring steps in the reward estimate

**Strengths:**

- the paper is well written and organized, with rich benchmarking experiments and comparisons
- strong experimental results in comparison to baselines

**Weaknesses:**

- the novelty is incremental, focusing on two additions to vanilla OT
- the presentation of the central claim seems to contradict the rationale behind the two additions to OT (see the question below).
- some result analyses contain plain observations, lacking in-depth insights
	- missing discussion on why learning from demonstration performs worse in door-lock and window-open tasks (sec. 4.3), important for understanding the efficacy scope of the proposed extension; in contrast, RL with simply binary reward performs well in the twos
	- no insights on why important components vary across tasks (sec. 4.4), beneficial for understanding the sensitivity of the proposed extension
   - ablation of $k_c$ and $k_m$ in sec. 4.4 is good, but a high-level guide for choosing them under different task conditions would benefit others following this work and be important for understanding the generalisation capability of the two additions
   - as to sec. 4.6, the description of the pixel-based setting is a bit over-simplified, the author can instead put detailed explanations in the appendix
   - sec. 4.7 feels like a case-by-case study; it does not relate to the method's efficacy and the paper's central claim

- minor comment:
     - a brief description of the nine meta-world tasks (and task length) would provide an overview of task complexity
     - a detailed example of computing group-wise cosine similarity is missing. Including this would help clarify the value of introducing $k_c$ in the cost estimate, especially concerning the example on the right side of Figure 4

**Questions:**

### Presentation of the central claim
- in section 3.1.2, how does the permutation study with ADS connect to the first observation; is being oder-invariant within a trajectory problematic? Which proposed component (masking scheme or context embeddings) addresses this? Further, how does the 1st observation relate to the claim that existing methods overlook temporal information (e.g., lines 44-45)?
- what exactly is meant by *overlooking temporal information*? eq. 5 already counts all future steps in calculating the original OT proxy reward for current state. Why incorporate temporal information?
- the paper claims to incorporate this information (lines 144-145), but doesn't the masking scheme *weaken* the influence of future/distant steps (lines 149-152), while context embedding *enlarges* it (lines 168-169)? This divergence causes confusion about the central claim of mitigating the issue of overlooking temporal information (e.g., lines 8-12, 45-47)
	- Introducing context embeddings (section 3.2.2) seems to contradict the second observation in section 3.1.2 (lines 116-117) as considering later steps could add noise to the reward estimate of the current state-action pair.

### Methodology
- is the context cost matrix in eq. 9 used in the optimisation of transport plan $\mu$ in eq 4, or just for reward inference in eq. 5?

### Experimental results
- in Figure 6, the average of TemporalOT appears less than 7.5 but is reported to be 8.4 in Table 1.
- in Figure 2, which step does $o_2$ correspond to? better to add x-y labels and mark $o_2$, $o_3$ to the reward curve on the right
- misspecified value for TaskReward in Door-lock env, according to Figure 9, the mean value is around 60. Similarly, the Window-open env mean value should be around 30
- which $\epsilon$ values were used in the experiments (or are they determined after optimizing equation 7), and how do they affect the task performance?
- the 4th baseline OT needs more explanation, though I can roughly guess how online OT differ from offline OT

### Discrepancy in ASD vs. OT performance

- unsure about why ASD performs worse than vanilla OT in some tasks as shown in Table 1? I understand this paper adopts a more challenging experimental setup, with varying goals per episode and only two demonstration videos provided. However, in the benchmarking results of the ASD paper, ASD achieves over 80% success in the basketball task within 1M steps, significantly higher than OT's performance (around 10%), as shown in Figure 11 of Appendix C.2. This trend is also observed in the lever-pull task. Yet, Table 1 in this paper shows the opposite comparison between ASD and vanilla OT for these tasks. Could you explain this discrepancy?

### Minor points

- Figure 1, five steps should be -> six steps? the scheme is not indicative of the magnitude of reward $r_a$ or $r_b$; what does $a_2^a$ differ from $a_2^b$?
- It would be better to indicate in the caption of Table 1 that the values represent success rates, even though this is mentioned in section 4.1, as tables should be self-contained

**Limitations:**

- authors acknowledge the limitations of the work, and their comments are fair
- it's unclear how different pretrained visual encoders will influence OT's efficacy under the same conditions; a discussion on this topic would be helpful
- nonetheless, the authors discuss the broader impact of their work, including immediate societal implications, and suggest a straightforward solution, which is appreciated

---

> ### Author Rebuttal · Authors · 2024-08-02
>
> We thank the reviewer for the detailed and constructive comments.
>
> ## Questions
>
> > **Q1: Presentation of the central claim**
>
> Due to the importance of this point and limited space, we have clarified the motivation/central claim in the common reply (top) on Clarifications on motivation.
> In short, the permutation study provides empirical evidence that the conventional OT-reward (Eq 5) is order-invariant (1st obs),  since the frame order in the expert trajectory is fully discarded.
>
> > **Q2: What exactly is meant by overlooking temporal information? How does 1st obs relate to it**
>
> We meant overlooking the temporal *order* information (apologize and will revise). Since Eq 5 is order-invariant, methods using it will overlook order info. We further detailed on this in C1.
>
> > **Q3: Doesn't the masking scheme weaken ... while context embedding enlarges....**
>
> Both scheme are for incorporating the temporal *order* information (C2).
> This seems to be caused by the term "temporal information". We replaced it with "temporal order information" in the revised paper.
>
> > **Q4: Introducing context embeddings seems to contradict the second observation**
>
> The context length $k_c$ (i.e., 2-3) is much smaller than the trajectory length (i.e.,100-200). In observation 2, we discuss the distraction from distant states (i.e.,100-200 steps). The context embedding with a small context length $k_c$ will have a small influence on the distraction issue as discussed in observation 2.
>
> > **Q5: Is the context cost matrix in Eq9 used in the optimization of transport plan 𝜇 in Eq4, or just for reward inference in Eq5?**
>
> The context cost matrix in Eq9 is used in *both* transport plan optimization and reward inference.
>
> > **Q6: In Figure 6, the average of TemporalOT appears less than 7.5 but is reported to be 8.4 in Table 1**
>
> This is because we plot a smoothed evaluation curve. Table 1 reports the result at the last step: (7+7+11+10+7)/5 = 8.4. Figure 6 plots a smoothed curve: (5+4.33+8+9.33+7)/5 = 6.73. Logs are available in PDF-Table 1 in the uploaded PDF.
>
> > **Q7:  $o_2$ in Figure 2**
>
> $o_2$ is the 40th step.
>
> > **Q8: Misspecified value for TaskReward**
>
> Sorry that we made a mistake when plotting Fig 9. We extracted the same number of data samples for TaskReward from the evaluation log as other baselines, and thought it corresponds to1e6 steps. But actually it is 5e5 steps since it uses doubled evaluation frequency. PDF-Fig 7 is the corrected figure matching Table 1
>
> > **Q9: Effect of 𝜖**
>
> We follow the parameter setting from ADS where 𝜖=0.01. PDF-Fig 6 from the PDF file shows the ablation for different 𝜖. The entropy coefficient 𝜖 controls the sparseness of the transport plan 𝜇. A large value will generate a 𝜇 closer to uniform distribution.
>
>
> > **Q10: Discrepancy: Table 1 shows the opposite comparison between ASD and vanilla OT for basketball and lever-pull**
>
> Sorry about the confusion. OT in the ADS paper corresponds to OT0.99 in Table 1 of our paper. By comparing OT0.99 with ADS, we can see that the results are *not opposite*, with ADS outperforming OT0.99 on basketball and lever-pull tasks. The overall performance drop is indeed due to the more challenging experimental setup as you mentioned.
>
>
> > **Q11: Figure 1, five steps ... what does $a^a_2$ differ from $a^b_2$**
>
> Sorry that *steps* is a confusing term and we changed it to *steps of transitions*.
> $r_b$ is larger than $r_a$ since  $a^b_2$ leads to a success grasp ($a^a_2$ fails resp.) and matches demo better.
>
> Thanks for all the good suggestions and will include the improved figure in revision.
>
>
> > **Q13: Different pretrained visual encoders**
>
> Thanks for the suggestion and we will add more discussions in revision.
>
> In this work, we follow the standard setting [ADS] and use a pre-trained ResNet50 to extract the visual embedding.
> Other visual encoders could also be used. We also run some experiments quickly to compare different visual encoders in PDF-Fig 2 in the uploaded PDF file. In PDF-Fig 2, we compare three different ResNet variants (ResNet18, ResNet50, ResNet152). PDF-Fig 2 indicates that a reasonably good visual encoder is usually enough to extract effective visual embeddings for computing OT rewards in RL.
>
>
> ## Weaknesses
>
> > **W1: Novelty**
>
> To the best of our knowledge, we are the first to discuss how to leverage temporal order information in computing OT rewards for training an RL agent. As an initial attempt on this topic, we aim to address this problem from a minimalist perspective. Though being simple, the method is effective and could be a good starting point to inspire others to reproduce and build upon.
>
> > **W2: Bad performance in door-lock and window-open**
>
> Because these two tasks have strict success conditions, i.e., the door-lock task is defined as solved only when the knob rotates to 60°. IL proxy reward is not accurate enough for such details, i.e., 55° w.r.t. 60°.
>
> > **W3: Important components vary across tasks**
>
> When long distance distraction is the main bottleneck, temporal mask is the key. When inaccurate transport cost is the main bottleneck, context embedding is the key.
>
> > **W4: How to choose of $k_c$ and $k_m$**
>
> A small value of $k_c$=3,4 is effective enough for a context information-augmented OT cost, while a larger value will oversmooth the cost information. $k_m$ relates to the length of the expert trajectory L. Setting $k_m$ to around 0.1L is a good rule of thumb.
>
> > **W5: Example of computing group-wise cosine similarity**
>
> In Figure 4, we use $k_c=3$. The transport cost between $o_1$ and $o^E_2$ is: $\hat c(o_1, o^E_2) = 1 - \left[\mathrm{cos}(f(o_1),f(o^E_2))+\mathrm{cos}(f(o_2),f(o^E_3))+\mathrm{cos}(f(o_3),f(o^E_4))\right]/3$.
>
>
> ## Other
> > **Caption of Tab 1; details about sec 4.6 in appendix; comments on sec 4.7; description of meta-world; more explanation on the 4th baseline ...**
>
> Thanks for all the other comments (cannot list all due to space; all received, thanks!) and we have revised paper accordingly.

---

> > ### Comment · Reviewer_GZSS · 2024-08-11
> >
> > Many thanks to the authors for thoroughly addressing my concerns and adding the extra experimental plots, much appreciated. I found the clarifications on C2 helpful and suggest (if possible) incorporating them into sec. 3 when introducing new techniques. It would also be great if you could include the insights from W3 and W4 in the discussion section.
> >
> > I have just a few other points:
> >
> > - Is the context cost matrix in eq. 9 used in optimizing the transport plan $\mu$ in the new objective (eq. 5)? It seems NO as shown in Fig. 4. If yes, shouldn’t the formulation of eq. 6 and 7 reflect the context cost matrix $\hat{C}$?
> >
> > - Regarding W2, you mention that the "IL proxy reward is not accurate enough for such details." How is this generally true for imitation learning (IL)?

---

> ### Author Response · Authors · 2024-08-12
> **To Reviewer GZSS: Explanations for Figure 4 and proxy reward**
>
> > **S1: I found the clarifications on C2 helpful ... It would also be great if you could include the insights from W3 and W4 in the discussion section...**
>
> It is great to know that we have successfully resolved the confusions and addressed most of the questions you had, and only a few other points need discussions (as done below).
>
> We greatly appreciate many of the great points you raised that help to improve the quality of the paper (e.g. the clarifications on C2 and insights from W3 and W4). We will definitely include them in the revised paper (e.g. into Sec. 3 and discussion section respectively).
>
> > **S2: Cost Matrix in Figure 4**
>
> The context cost matrix is used in optimizing the transport plan 𝜇.
>
> We apologize for the confusion. After carefully understanding the point you raised, we realized that it is caused by our order of introducing the two components of TemporalOT.
>
> Our original intention was to organize Section 3.2.1 and Section 3.2.2 to introduce the two components of TemporalOT in a progressive way starting from vanilla OT: firstly introducing temporal mask M in Section 3.2.1, and then the component for improving the cost matrix calculation in Section 3.2.2. Figure 4 in the main paper is also organized under the same reasoning.
>
> Thanks to your reminder, now we realized that this might introduce some confusions if we want to understand the whole method from the first section only (e.g. Eqn.(6) and Eqn.(7)) if there is no explicit reminder around  Eqn.(6) and Eqn.(7) (or Figure 4 Left) saying the $C$ in Eqn.(6) and Eqn.(7) (respectively Figure 4 Left) will be upgraded with a context embedded version $\hat C$ later.
>
> We are planning to introduce the Context Embedding based Cost Matrix (Eqn.(9)) in the original paper before Eqn.(6). Figure 4 will also be updated accordingly: 1) introduce Context Embedding on the left and then 2)
> temporal mask based OT on the right with the Cost Matrix notation $C$ changed to $\hat C$.
>
> We believe this can resolve the confusion as pointed by you. Thanks again and we will also include this change into the revision together with the ones as mentioned in reply to **Suggestion 1 (S1)**.
>
> We would thanks the reviewer sincerely again for all the great advice that helps to further improving the presentation quality of the paper.
>
> > **S3-1. IL proxy reward is not accurate enough for such details**
>
> Thank you for the question.
>
> We are sorry for the inaccurate and confusing "IL" term. By "IL proxy reward", we actually meant "visual similarity-based reward", i.e. reward calculated based on the visual similarity between the policy trajectory and reference trajectories. The term "IL" term is really unnecessary here, sorry for the confusion.
>
> It is easy to understand that due to factors such as viewpoint, occlusion, object sizes etc, sometimes it can be challenging to differentiate subtle differences in images, e.g., the knob being turned to 55° (task failure) vs a knob being turned to 60° (task success). While there is a clear difference if a low-dimensional knob angle sensor is provided (which is the case in low-dimensional state based reward), the visual signal based reward does not have the luxury of accessing the knob angles directly, leading to the challenges as mentioned above.
>
> > **S3-2: How is this generally true for imitation learning (IL)**
>
> Sorry for the usage of the confusing "IL" terms as explained above. It was an inaccurate term we used to describe the visual similarity based reward and we didn't intent to pass any implication on the Imitation Learning in our original response.
> After clarifying all the confusions, we can provide some of our understanding on visual observation based imitation learning. Based on our understanding, some of the challenges we observed in visual-similarity based reward calculation could also appear in imitation learning in several cases:
>
> - Imitation-learning with visual reward: this is the category that aligns with the line of works in ADS etc and this paper, which actually formulate IL into an RL form with the help of the visual similarity reward. This is easy to understand since the visual-reward calculation is directly impacted as explained above.
>
> - Visual BC-type of imitation learning: behavior cloning (BC)-type of IL methods does not require the proxy reward so the reward part is not an issue. However, if visual observation is used as the policy input, differentiating subtle differences based on visual observations is similarly challenging and will likely to impact policy learning in our opinion.
>
> Moving forward, general techniques that can improve on these aspects can potentially improve both the visual-similarity based proxy reward (the line of works in ADS etc and in this paper) and visual similarity-based imitation learning. This is a very interesting direction for future work, with potential impacts on multiple fields.

---

> ### Author Response · Authors · 2024-08-13
> **To Reviewer GZSS: A gentle reminder**
>
> Dear Reviewer GZSS,
>
> As we are approaching the end of the rebuttal period, we want to send a gentle message to you confirming that our reply to your followup question on "Figure 4 and proxy reward" has been successfully received (it seems that there was an issue with the OpenReview system, as it did not send out notification emails for some reason when we posting our replies, and we didn't receive any email notifications after we posted the replies; so we send this message just in case it also happened to you),  and we also want to confirm the reply has addressed your followup questions on "Figure 4 and proxy reward".
>
> Looking forward to your confirmation and thanks again for your time and efforts in reviewing our paper and providing detailed, constructive comments.
>
> We greatly appreciate many of the great points you raised that help to improve the quality of the paper, and we enjoyed the interaction with you during the discussion period.
>
> We will definitely incorporate the valuable clarifications and insights formed in our discussion into the revised paper.

---

> > ### Comment · Reviewer_GZSS · 2024-08-14
> >
> > Thanks to the authors for their responses, which clear up my follow-up questions and offer details on visual-similarity-based reward calculation and imitation learning. I appreciate the added clarity.

---

### Official Review · Reviewer_M6BJ · 2024-07-11

**Soundness:** 3
**Presentation:** 3
**Contribution:** 2
**Rating:** 5
**Confidence:** 4

**Summary:**

This work proposes a reinforcement learning method from a few expert demonstrations based on optimal transport. It incorporates temporal information into the framework of optimal transport reward so that the agent can focus on more relevant information in learning. The work proposes two simple tricks to achieve this, i.e., a masking mechanism and context embedding-based cost matrix. The experiments in simulated robotic control benchmarks show that the proposed temporal OT converges to a higher success rate than imitation learning and inverse reinforcement learning methods that do not consider temporal information.

**Strengths:**

The paper is clearly motivated. It points out a major weakness of OT-based reward calculation, that is the temporal information not being considered, and proposes a simple solution that applies a temporal mask to the cost matrix.

**Weaknesses:**

1. TemporalOT uses a manually designed diagonal-like matrix as the mask on the transportation plan, which could be ineffective when the learning policy has not been close to the expert or gets stuck at early-stage behaviors. Have the authors tried other design choices for the temporal mask or tried to make it a part of the learning?
2. As a robot learning paper, it would be better to apply the algorithm to real robots similar to previous OT-based methods (e.g., [18]).

**Questions:**

The variance of the baseline ADS is quite large in some environments shown in Fig. 7. Is there any explanation/discussion on it?

**Limitations:**

As shown in the learning curves, the proposed algorithm requires ~1e6 timesteps to converge, which may limit its practical use for real robots. I think initializing the policy with imitation learning, then tuning it with TemporalOT might converge faster.

---

> ### Author Rebuttal · Authors · 2024-08-05
>
> We thank the reviewer for the detailed, constructive review and insightful comments.
>
> Your valuable comments helped us to further improve the quality of the paper.
>
> Responses to the questions are below:
>
> > **Q1: Try other design choices for the temporal mask or try to make it a part of the learning**
>
> Thanks for the great question.
>
> We used the current design in the paper is because:
>
> - We want to use a simple and easy to understand design to illustrate the benefits of incorporating temporal information into OT rewards;
> - The current design is interpretable, easy to implement and computationally cheap.
>
> To mitigate the issue that the agent could get stuck at early-stage behaviors, here, we introduce a variant of TemporalOT that uses a dynamic mask for each trajectory. We begin with some notations:
> - The agent trajectory is $\tau = (o_1, \cdots, o_T)$ and the expert trajectory is $\tau^E = (o^E_1, \cdots, o^E_T)$.
> - We compute a cosine similarity based transport cost matrix $\hat C\in\mathbb R^{T \times T}$ as Eq 9.
> - For each step $o_i$ in the agent trajectory, we compute a mask window $M_i\in\mathbb R^{1 \times T}$ where $M_i(j)$=0 or 1 for $1\le j \le L$.
>
> In the current paper, we use a variant of the diagonal matrix with a mask window size $k_m$:
> $$
> M_i(j) = \begin{cases}1, &\text{if } j \in [i-k_m, i+k_m] \\\\0, &\mathrm{otherwise} \end{cases}
> $$
>
> In the added experiment, we learn a dynamic mask window for each step as follows:
> $$
> M_i(j) = \begin{cases}1, &\text{if } j \in [c-k_m, c+k_m] \\\\0, &\mathrm{otherwise} \end{cases}
> $$
> Here, we select the window center index as $c = \arg\min_j \hat C(i, j), j \in [0, i]$. This means we select an index $j$ in the expert trajectory that has the least transport cost w.r.t $o_i$. We further add a constraint $j \in [0, i]$ to avoid looking into distant future steps as pointed out in Section 3.1.2.
>
> The experiment results are shown in PDF-Fig 4 of the attached PDF. We can observe that this learning-based temporal mask slightly outperforms our previous rule-based temporal mask.
>
> We have been focusing on using a simple and effective approach to illustrate the main point of the paper, and therefore did not try to learn the temporal masks. We do agree with your intuition that learning temporal masks is a very promising direction to pursue.
>
> > **Q2: As a robot learning paper, it would be better to apply the algorithm to real robots similar to previous OT-based methods**
>
> Thanks for the suggestion.
>
> We apologize that simulated environments are mainly used in experiments.
> The reason is that we currently have no real robots at hand.
> At the same time, in our humble opinion, benchmarking different methods under the same setting in simulation also provide valuable empirical verifications on the effectiveness of different methods.
>
> We fully agree with you that it is important to validate the effectiveness of the proposed method in real-world scenarios.
> We are very willing to deploy our algorithm to real robots once physical robots are available in the future.
>
> > **Q3: High variance of ADS in some environments.**
>
> Thanks for the great question.
>
> Actually, the ADS baseline also shows high variance in its original paper [ADS]. The main reasons are:
>
> - ADS uses the naive OT reward definition (Eq 5) that depends on the whole trajectory. Such a global scope sometimes introduces noisy information from far away steps. Therefore, we can observe that both the ADS and OT show high variances in Figure 4, 5, 10, 11, 12 from the ADS original paper.
>
> - Moreover, the discount factor changes during the training in ADS. This makes the TD target less stable. Sometimes, such a changing discount factor will lead to a change in the Q-value function and the learned policy. In some tasks, the less stable ADS policy will also lead to high variance.
>
>
> > **Q4: As shown in the learning curves, the proposed algorithm requires ~1e6 timesteps to converge, which may limit its practical use for real robots. I think initializing the policy with imitation learning, then tuning it with TemporalOT might converge faster.**
>
> Thanks for the question.
>
> Using imitation learning to initialize the robot policy and then fine-tuning it with TemporalOT will very likely to help converge faster.
>
> One thing to note is that the demos used in our work and previous literature [e.g. ADS] are assumed to be **action-free videos**. Therefore, many of the standard action-based imitation learning approaches are not applicable, preventing a simple imitation-based policy pre-training.
>
> To do policy pre-training, state-based imitation learning approach should be used. However, the proposed approach is actually a type of state-based policy learning approach already. From this sense, although we can further improve sample efficiency with pre-training, it can be regarded as a factor that can be used by all baselines. Therefore, for fair comparison in this work, we place all the baseline methods on the same footing without policy-pre-training.
>
> On the other hand, to verify the point raised in your comment and go beyond the settings used in this work, we did some quick experiments with **action-inclusive demonstrations**. In PDF-Fig 5 of the attached PDF file, the BC Pretrain baseline refers to pre-training the policy with imitation learning. In addition, the BC Loss baseline refers to using both imitation learning loss and RL-loss for policy training throughout.
> We can observe that both BC Pretrain and BC Loss show performance gain in terms of sample efficiency, which confirms your intuition.
>
> **References**
>
> [ADS] Liu et al., Imitation Learning from Observation with Automatic Discount Scheduling, ICLR 2024

---

> > ### Comment · Reviewer_M6BJ · 2024-08-11
> >
> > Thank you for the rebuttal. I appreciate the added results regarding other design choices of temporal masks.
> > I have some follow-up questions:
> > In PDF-Fig 5, why does the curve for "BC pretrain" start from a success rate of 0 instead of a positive success rate learned by BC? I am asking the BC and fine-tuning results to see how many offline demonstrations and how many online interactions are required to obtain a reasonable success rate, and to justify how practical the algorithm is for potential real-world applications.

---

> ### Author Response · Authors · 2024-08-12
> **To Reviewer M6BJ: Explanation for PDF-Fig 5**
>
> It is great to know that we have successfully addressed most of your questions. We appreciate your great efforts in reviewing our paper that helped to improve our work.
>
> As to your follow-up question on "BC pretrain" results in PDF-Fig 5 at step-0, we apologize that this was caused by a setting that was inappropriate for methods with pre-training in generating the plot. BC-pretrain has a success rate around 10.8 $\pm$ 9.2 at step-0 for Door-open task (consistent with the result reported in Table 1 of the main paper).
>
> What happened was when we generated PDF-Fig 5, the first step of actual evaluation happens at 2e4 steps which was set according to all the online learning methods (such as ADS and TemporalOT, which only start to learn after collecting some online samples via environmental interactions). A default success rate of zero is used since all these methods always have a zero success rate at the step-0 (essentially before any training).  The issue was that for generating PDF-Fig 5, we forgot to adapt the script for BC-pretrain method to run evaluation at step-0 and used a default success rate of zero. While this makes sense for the online learning methods (w.o. pre-training) as explained above, for BC-pretrain, it is inappropriate since it has a non-zero success rate before any online training because of the pre-training.
>
> Also in order to reflect the original values, we didn't apply smoothing in PDF-Fig 5, therefore the curve values remains at zero at step 0 (some curves in Figs elsewhere e.g. main paper with minor non-zero values at step 0 are due to smoothing, and the raw values are zero).
>
> To resolve the confusions and maintain consistency in presenting the results, we are planning to change PDF-Fig 5 in to a Table as shown below to be included in our revised paper (due to the reason that while Figures are good for comparing the general trends across the training period, Table is more suitable in this case since it directly shows the raw results at the actual evaluation steps and makes it easier to to compare the pre-train-only performance and the pre-train+online training performance).
>
> |                                                   | 0           | 2e4        | 4e4          | 6e4          | 8e4          | 1e5          | 5e5          | 1e6          |
> | ------------------------------------------------- | ----------- | ---------- | ------------ | ------------ | ------------ | ------------ | ------------ | ------------ |
> | BC [pure-offline, param fixed after pre-training] | 10.8 $\pm$ 9.2 | -          | -            | -            | -            | -            | -            | -            |
> | TemporalOT [pure online]                          | 0.0 $\pm$ 0.0  | 2.0 $\pm$ 4.0 | 0.0 $\pm$ 0.0   | 0.0 $\pm$ 0.0   | 5.0 $\pm$ 10.0  | 16.6 $\pm$ 32.2 | 57.8 $\pm$ 37.1 | 78.4 $\pm$ 12.4 |
> | TemporalOT with BC-pretrain [offline-to-online]                      | 10.8 $\pm$ 9.2 | 6.8 $\pm$ 8.4 | 25.0 $\pm$ 18.5 | 42.8 $\pm$ 22.2 | 48.6 $\pm$ 25.9 | 55.4 $\pm$ 12.0 | 70.8 $\pm$ 8.2  | 82.0 $\pm$ 2.4  |
>
>
> We can observe that:
>   1. BC without any online training has relatively low success rate (10.8 $\pm$ 9.2)
>   2. TemporalOT achieved relatively high success rate after 1e6 environmental steps (78.4 $\pm$ 12.4)
>   3. Incorporating pretraining (Temporal OT with BC-pretrain) helps to further improve the sample efficiency. We want to clarify as already done in the original rebuttal reply that our default setting in the main paper is an action-free-demo setting. For this ablation study, we go beyond this setting and use action labels which are required by BC.
>   4. For TemporalOT with BC-pretrain, one can notice a small success rate drop (which recovers later) at the initial phase when we transit from offline to online training (step0->step2e4->step4e4). This is an "initial-dipping" phenomenon as also being reported in previous literature and can be improved by designing more specific offline-to-online methods [O2O, PEX etc].
>
> In summary, incorporating offline pre-training into TemporalOT does show promising improvements to further boost the sample efficiency and indicate practical paths to generalizing and applying TemporalOT in real world applications (thanks for sharing your insights with us on this!). And we believe the offline-to-online training paradigm in general [O2O, PEX etc], although out-of-the scope of the current paper (which only focus on the online training setting) is a powerful one for real-world robotics applications and we plan to pursue it in our future work.
>
> **References**
>
> [O2O] Yu et al. Actor-Critic Alignment for Offline-to-Online Reinforcement Learning, ICML 2023
>
> [PEX] Zhang et al. Policy Expansion for Bridging Offline-to-Online Reinforcement Learning, ICLR 2023

---

> > ### Comment · Reviewer_M6BJ · 2024-08-13
> >
> > Thank you for the additional explanation. My questions are addressed and I will raise the score to 5.

---

> > > ### Author Response · Authors · 2024-08-13
> > > **To Reviewer M6BJ: Thank you for your positive feedback and for raising the score**
> > >
> > > We thank Reviewer M6BJ for your positive feedback on the rebuttal and thank you for raising the score. We greatly appreciate your efforts in reviewing the paper and your valuable comments that have helped to further improve the quality of the paper.

---

### Official Review · Reviewer_FnLe · 2024-07-11

**Soundness:** 3
**Presentation:** 3
**Contribution:** 3
**Rating:** 6
**Confidence:** 4

**Summary:**

This paper proposes several improvements on top of the Optimal Transport reward for inverse reinforcement learning. The key observation is that the traditional OT-based reward does not consider the temporal information, i.e., it is invaraint to the order of the state-action pairs in a trajectory, and that the reward is not defined over state-action pairs but over a whole trajectory, so later transitions in the trajectory would affect the reward of an early state-action pair, which does not follow the standard definiation of reward in RL. This the key technique the paper proposes is to introduce a mask for the transport plan, such that only temporaily neighboring states between expert and the agent will be considered when computing the reward. It futher proposes a technique to futher improve the computation of the cost matrix in the OT reward. Experiments are conducted on 9 metaworld tasks, and the proposed masked OT rewards outperform several previous baselines. Ablation studies are also conducted to help understand some design choices.

**Strengths:**

- This paper studies an important problem in the formulation of the OT-based reward, which is it being non-temporal (invariant to the order of the trajectory).
- The paper is well-written and easy to follow.
- The proposed algorithm is simple yet effective, and it achieves better performances than prior methods in the experiments.
- The experiments are thorough, with results compared on 9 environments, and with ablaiton studies that are very helpful for understanding how each component helped to improve the performance.

**Weaknesses:**

- The proposed method to restrict the temporal information works by masking the transport plan to only include temporaily nearby states. This "temporaily nearby" is measured using the distance between the time step indexes, which assumes that the time interval between each step, or the time scale, is similar between the agent and the expert. I am wondering how general this is true. I encourage the authors to think of cases where this assumption might be broken and add some discussion on how the method would work & can be extended to work in such cases.

**Questions:**

Please refer to the weakness section.

**Limitations:**

Yes.

---

> ### Author Rebuttal · Authors · 2024-08-04
>
> We thank the reviewer for your positive review and insightful comments!
>
> Your valuable comments helped us to further improve the quality of the paper.
>
> Responses to the questions are below:
>
> | **Q: The proposed method to restrict the temporal information works by masking the transport plan to only include temporarily nearby states. This "temporarily nearby" is measured using the distance between the time step indexes, which assumes that the time interval between each step, or the time scale, is similar between the agent and the expert. I am wondering how general this is true. I encourage the authors to think of cases where this assumption might be broken and add some discussion on how the method would work & can be extended to work in such cases.**
>
> Thanks for the great question.
>
> Our work indeed inherits an implicit assumption from *learning from demonstration* literature:
>
> - The agent has a similar movement speed as the expert.
>
> Under this assumption, using the distance between the time step indexes is a simple and effective strategy to represent temporal affinity information.  We will make this clear in our revised paper.
>
>
> As to your question on wondering how general this is true, we do admit that this assumption can be broken if the discrepancy between expert-agent movement speed / trajectory alignment is large enough.
>
> Here we share some quick empirical results that we were able to finish within the limited time on investigating the performance v.s. discrepancy (more results with the complete transition phase, i.e. the stage that it starts to fail due to increased discrepancy, will be added to the appendix of revised paper).
>
> We have done experiments to use double speed expert demonstrations, where we get the demos by sampling the original expert trajectory every 2 steps. PDF-Fig 3 in the uploaded PDF file shows the experiment results. Though the expert trajectory length is only half of the agent trajectory length, the proposed TemporalOT method is still effective under a such setting. This experiment provides some positive empirical evidence that there are some tolerance in terms of the deviation from the assumption (discrepancy).
>
> In addition to these empirical investigations, as a future extension, we may need to further generalize the definition of temporal affinity accordingly, potentially at a more abstract level. For example, we can first learn to infer the intention/latent goal of the agent and then compute the OT reward w.r.t. the intentions [Ghosh et al., 2023]. This is a very interesting direction that is worth exploring in future work.
>
>
> **References**
>
> [Ghosh et al., 2023] Reinforcement learning from passive data via latent intentions, ICML 2023

---

> > ### Comment · Reviewer_FnLe · 2024-08-10
> > **Official Comment**
> >
> > I want to thank the authors for the detailed and well-written response, and for adding a new experiment within the short rebuttal period of time. I don't have further questions and I remain positive about the paper.

---

> ### Author Response · Authors · 2024-08-10
> **To Reviewer FnLe: Thank you for your positive feedback**
>
> We thank Reviewer FnLe for your positive feedback on the rebuttal. We greatly appreciate your efforts in reviewing the paper and your valuable comments that have helped to further improve the quality of the paper.

---

### Official Review · Reviewer_hYj8 · 2024-07-30

**Soundness:** 3
**Presentation:** 3
**Contribution:** 3
**Rating:** 7
**Confidence:** 3

**Summary:**

The authors introduce TemporalOT, a learning-based proxy reward that incorporates temporal information. via using a mask mechamism and context-embeddings based cost matrix. They implemented the TemporalOT-RL based on ADS implementation, conducted thorough experiments on Meta-world benchmark focusing on the a challenge setting where there are only two demonstrations available and outperformed the OT-based reward baseline.

**Strengths:**

1. The motivation of incorporating temporal information into OT-reward is clear.
2. The paper is well-written and easy to follow.
3. The experiments are thorough and convincing
4. The ablation study of no-mask, no-context, discount factor, number of demonstrations is very sufficient.

**Weaknesses:**

See Questions

**Questions:**

1. Could the author analyze what potential property of the push task makes it different that ADS outperforms TemporalOT?
2. Does the author intend to incorporate visual encoder training instead of fixed pre-trained?

**Limitations:**

Performance relies heavily on the demonstration quality and visual encoder effectiveness.

---

> ### Author Rebuttal · Authors · 2024-08-03
>
> We would like to thank the reviewer for the positive review and for the efforts in reviewing our work!
>
> Your comments are valuable in helping us to further improve the quality of the paper.
>
> Responses to the questions are below:
>
> | **Q1: Could the author analyze what potential property of the push task makes it different that ADS outperforms TemporalOT?**
>
> Thanks for raising this point. The goal of the push task is to move the red cylinder to a target position (green dot) on the surface of the table, as illustrated in Figure10 (C) in the appendix. Usually, we can solve a task by focusing on the robot arm trajectory in the demo, i.e., open a door. However, we need to pay special attention to the red cylinder in the *push task* and move it to the exact target position.
>
> ADS performs better in this task because:
> - Firstly, the red cylinder is quite small compared to the red robot arm. Therefore, the cosine similarity based OT reward usually contains more information about the robot arm than the red cylinder. The agent is likely to focus on imitating the robot arm movement and ignores the small red cylinder.
> - Secondly, ADS uses a larger 𝛾 than TemporalOT due to its adaptive discount scheduler. From the comparison of OT0.99 and OT0.9 in Table 1 of the main paper, we can observe that using a larger discount factor indeed helps to improve performance.
>
> The benefit of using the adaptive discount scheduler in ADS is significant for the *push task* because the larger 𝛾 encourages the RL agent to look into more future steps via TD-learning and assign higher weight for future rewards. This is important because the OT reward is generally more noisy in the *push task* than the other tasks due to the small size of the red cylinder. For example, the OT reward $r_t$ at step $t$ is sometimes not that accurate but the OT reward $r_{t+3}$ at step $t+3$ is more accurate and contains more useful information about the red cylinder. Under a such circumstance, a larger 𝛾 will let the agent pay more attention to $r_{t+3}$ and learn a more accurate Q-value function and policy accordingly.
>
> In the experiment, we always use 𝛾=0.9 to match the initial 𝛾 value in ADS and the second best baseline OT0.9. We added experiments to run TemporalOT with a larger discount factor 𝛾=0.99. From PDF-Fig 1 in the uploaded PDF file, we can observe that TemporalOT with a larger discount factor achieves better performance in the push task.
>
> | **Q2: Does the author intend to incorporate visual encoder training instead of fixed pre-trained visual encoder?**
>
> Thanks for the great question. The reason for using a fixed pre-trained visual encoder in our work is mainly following standard settings in literature [e.g. ADS paper], which removes the factor of visual encoder learning and helps us to focus on reward structure design.
>
> Although it is not the main focus of this work, we can definitely incorporate visual encoder training instead of using a fixed pre-trained visual encoder. For example, we have the following different strategies to train the encoder:
>
> - In the pixel-based setting, we unfreeze the encoder in the critic and use the Q-learning loss to optimize the encoder.
> - We can also adopt some existing representation learning methods for RL to train the encoder, i.e., ATC [Stooke et al., 2021].
>
> Intuitively, this has the potential to further boost overall performance by reducing the potential domain shift issue, better capturing task-relevant signals, etc. Since the main focus of our current work is to demonstrate the effectiveness of adding temporal information to the OT rewards in RL, we follow standard setting in literature in order to make the comparison clear and consistent and leave the incorporation of visual encoder training as an interesting future work.
>
> | **Q3: Rely on demonstration quality and visual encoder effectiveness.**
>
> Thanks for the comment.
>
> **Demonstration quality.** We agree with you that this is a common limitation of imitation learning (IL). Since our method is closely related to IL, TemporalOT also faces this common issue. On the other hand, unlike many other IL methods that require a large diverse dataset of high-quality demonstrations, TemporalOT is designed to work with only a few expert demonstrations. For example, we only provide TemporalOT with two expert trajectories in the experiments. Such a small number of expert demonstrations makes our method more practical than the other IL methods, which require more expert demonstrations.
>
> **Visual encoder.** We leverage the visual encoder to extract embedding for each image and compute the cosine similarity based transport cost. Since this is also a common component shared by baseline methods such as OT and ADS,  any reasonably effective visual encoder can be used to make the comparison fair. In this work, we follow the experiment setup in ADS and use a pre-trained ResNet50 to extract the visual embedding.
>
> Other visual encoders could also be used. Following your suggestion, we also run some experiments quickly to compare different visual encoders in PDF-Fig 2 in the uploaded PDF file. In PDF-Fig 2, we compare three different ResNet variants (ResNet18, ResNet50, ResNet152) using the checkpoints from torchvision. We can observe that ResNet50 and ResNet152 encoders show similar final performances. Here, ResNet18 underperforms the other two encoders because it is quite weak, such that sometimes it fails to capture the key information of the image. PDF-Fig 2 indicates that a reasonably good visual encoder is usually enough to extract effective visual embeddings for computing OT rewards in RL.
>
> Given the rapid development of computer vision models, we can easily build our own model with some strong open-sourced vision encoder checkpoints. Moreover, we can incorporate visual encoder training, as discussed in Q2, to further improve performance.
>
> **References**
>
> [Stooke et al., 2021] Decoupling Representation Learning from Reinforcement Learning, ICML 2021

---

### Author Rebuttal · Authors · 2024-08-03

# To all the reviewers: Thanks for the reviews and summary of key paper changes

We thank all the reviewers (**R1**-hYj8, **R2**-FnLe, **R3**-M6BJ and **R4**-GZSS) for their time and efforts in reviewing the paper. The reviewers appreciated that:
- The motivation is clear and the problem is important (**R1**-hYj8, **R2**-FnLe, **R3**-M6BJ, **R4**-GZSS).
- The paper is well-written and easy to follow (**R1**-hYj8, **R2**-FnLe, **R4**-GZSS).
- The proposed method is simple and effective (**R1**-hYj8, **R2**-FnLe, **R3**-M6BJ).
- The experiments are thorough (**R1**-hYj8, **R2**-FnLe, **R4**-GZSS).

We have further improved the paper according to the valuable suggestions from all reviewers. Below is a summary of the key changes made to the current paper. We:
- added experiments for $\gamma$=0.99 on the push task (**R1**-hYj8) (c.f. PDF-Fig 1)
- added experiments to compare different visual encoders (**R1**-hYj8, **R4**-GZSS) (c.f. PDF-Fig 2)
- added experiments to use double speed expert demonstrations (**R2**-FnLe) (c.f. PDF-Fig 3)
- added experiments to use different masks (**R3**-M6BJ) (c.f. PDF-Fig 4)
- added experiments to incorporate BC pre-trained policy (**R3**-M6BJ) (c.f. PDF-Fig 5)
- added ablations for $\epsilon$ parameter (**R4**-GZSS) (c.f. PDF-Fig 6)
- re-generated Figure 9 by fixing a plotting issue that caused a mis-match with the results in Table 1 (**R4**-GZSS) (c.f. PDF-Fig 7)
- added clarifications and discussions on the assumptions of TemporalOT (**R3**-M6BJ, **R4**-GZSS).
- added more detailed explanations for the experiment results (**R4**-GZSS).

We greatly appreciate the efforts the reviewers spent on helping to improve our work and we are deeply grateful for all the valuable suggestions. We have carefully addressed each comment and improved our work accordingly. We hope they are helpful to address the concerns the reviewers had previously. We would love to hear feedback from all the reviewers.

# Clarifications on motivation
The motivation of this work has been well-perceived by **R1**-hYj8, **R2**-FnLe and **R3**-M6BJ (e.g. quoted from their reviews respectively: **R1**-hYj8: "the motivation of incorporating temporal information into OT-reward is clear"; **R2**-FnLe: "this paper studies an important problem in the formulation of the OT-based reward, which is it being non-temporal (invariant to the order of the trajectory)";  **R3**-M6BJ: "the paper is clearly motivated. It points out a major weakness of OT-based reward calculation").

Here we try to make some further clarifications to help resolve any potential confusions **R4**-GZSS seems to have (we apologize to **R4**-GZSS for any potential confusions we caused and we will resolve them below and also in the revised paper). We decide to re-clarify the motivation here since it is one of the most crucial prerequisite to better understand the method and resolve many related questions and confusions. Another reason to list it here is that it could be useful to readers from a broader audience later after the reviews and rebuttals are made public.

> **C1: Is being order-invariant within a trajectory problematic? Why order information matters in OT-based RL?**

Temporal order information is very important and being oder-invariant within a trajectory is problematic.

More specifically, in the standard OT-reward (Eq. 5), the order information is discarded and the frames from the demo sequence are treated as *bag-of-temporally-collapsed* frames. In our view, collapsing the temporal axis drops arguably one of the most important characteristic features of temporal information.

More concretely, consider a demo trajectory of 𝜏1 = (S1, S1, S2), meaning first to stay at S1 and then move to S2, and we want to imitate this behavior. However, if order information is discarded as in standard OT-reward (Eq. 5), from the perspective of reward, there is no ability to differentiate between 𝜏1 and some other (undesired) trajectories, e.g. 𝜏2 = (S1, S2, S1) (first move to S2 and move back to S1).
Therefore, it is possible for the policy to converge to a solution that generates trajectories like 𝜏2, which is undesirable.

In summary, discarding the temporal ordering information in reward calculation makes the reward on top of it under-constrained, which in consequence increases the possibility of converging to undesired solutions. This motivates us to investigate how to address this issue by incorporating temporal ordering information.

> **C2: Motivation of the proposed method. Which proposed component (masking scheme or context embeddings) addresses the lacking of ordering issue in the standard OT-reward?**

Both masking scheme and context embeddings contribute to the final overall performance and are incorporated into different stages in the algorithmic pipeline.

As a brief recap, the pipeline of standard OT-based RL is as follows:

(Stage 1) Transport cost matrix => (Stage 2) OT reward => (Stage 3) RL training

**Temporal mask.** We leverage the mask mechanism to incorporate temporal affinity and therefore the order information. This is the component that is designed to address the lacking of ordering issue in the standard OT-reward explicitly (Stage 2).

**Context embedding.**  This is a component for improving the quality of the cost matrix (Stage 1) by incorporating nearby context information. This component contribute to the inclusion of temporal ordering implicitly (and at a coarser temporal scale), by the incorporating of a temporal window as context for cost calculation (e.g. frames outside of the temporal window will not be considered in cost calculation).

We realized that the roles of these two components might not be well clarified in the original submission. We apologize to **R4**-GZSS for any potential confusions we caused and will resolve them in the revised paper.

---

### Decision · Program_Chairs · 2024-09-25

**Decision:**

Accept (poster)

**Comment:**

This paper proposes a method for inverse RL based on optimal transport, building on top of prior work with a new masking mechanism. Reviewers found the paper well written and well motivated, with thorough experiments. While the differences from prior work are small, the paper presents strong empirical results on metaworld. The authors and reviewers had a thorough discussion, which cleared up many questions and concerns about the paper.

Given that all reviewers scores indicate that the paper should be accepted, I’m recommending that the paper be accepted. I encourage the authors to incorporate the rebuttal experiments and discussion into the final version of the paper.

Minor comments: In Table 1, please bold all entries that are within the standard error of the best method (e.g., some of the numbers in the ADS column should be bolded).